# Cold-start Recommendation by Personalized Embedding Region Elicitation

**Hieu Trung Nguyen**[1,3]        **Duy Nguyen**[1]        **Khoa Doan**[2]        **Viet Anh Nguyen**[3]

[1]VinAI Research
[2]College of Engineering & Computer Science, VinUniversity
[3]The Chinese University of Hong Kong

## Abstract

Rating elicitation is a success element for recommender systems to perform well at cold-starting, in which the systems need to recommend items to a newly arrived user with no prior knowledge about the user's preference. Existing elicitation methods employ a fixed set of items to learn the user's preference and then infer the users' preferences on the remaining items. Using a fixed seed set can limit the performance of the recommendation system since the seed set is unlikely optimal for all new users with potentially diverse preferences. This paper addresses this challenge using a 2-phase, personalized elicitation scheme. First, the elicitation scheme asks users to rate a small set of popular items in a "burn-in" phase. Second, it sequentially asks the user to rate adaptive items to refine the preference and the user's representation. Throughout the process, the system represents the user's embedding value not by a point estimate but by a region estimate. The value of information obtained by asking the user's rating on an item is quantified by the distance from the region center embedding space that contains with high confidence the true embedding value of the user. Finally, the recommendations are successively generated by considering the preference region of the user. We show that each subproblem in the elicitation scheme can be efficiently implemented. Further, we empirically demonstrate the effectiveness of the proposed method against existing rating-elicitation methods on several prominent datasets.

## 1 INTRODUCTION

The vast amount of available information and content in this digital era poses severe challenges for individuals seeking information and recommendations. Personalized recommender systems are exerting profound impacts on various fields by leveraging user data to generate personalized suggestions, with applications spanning from networks [Natarajan et al., 2020, Wu et al., 2019], e-commerce [Alamdari et al., 2020, Jiang et al., 2019] to e-learning [Khanal et al., 2020, George and Lal, 2019]. Personalized recommender systems may produce accurate suggestions for a user's preferences by exploiting users' characteristics and historical interactions with items. However, under the cold-start settings, recommendation models may fail miserably because they can only access a limited, or even no, user's interaction history [Gope and Jain, 2017]. Cold-start recommendation can be categorized into three main branches, depending on the specification of the user-item pool: (i) new users, invariant items, (ii) invariant users, new items, and (iii) new users, new items [Gope and Jain, 2017]. Introductory materials for the cold-start problems in recommendation systems can be found in Adomavicius and Tuzhilin [2005], Schafer et al. [2007], and extensive literature reviews can be found in Bobadilla et al. [2013], Gope and Jain [2017]. We will provide a brief literature review in Section 5.

This paper considers the most popular setting of cold-start recommendation wherein a new user arrives, and our goal is to make relevant recommendations to the user from the list of existing (invariant) items. We assume that there is no available side information regarding the new user, and thus, there are no trivial methods to initialize the representation of the user in the system. This situation arises frequently in practically any real-time environment, for example, when a new user signs up for a new account on an online platform.

Despite their effectiveness, existing methods select a fixed set of elicited items to infer a new user's preferences on the remaining items [Sepliarskaia et al., 2018]. They effectively approximate the initial embedding of the user as a function of only these seed items, no matter who the user is. This approach works well when the user has broad interests in multiple item categories but could be wasteful for users with narrower interests, resulting in sub-optimal initial

recommendations. In practice, one can easily see that most users prefer some specific categories of items; for example, a user may be interested in action, adventurous, and mystery movies but not others such as comedy, drama, or romance. Thus, dynamically selecting or personalizing the seed items for a new user to represent the user's initial embedding can be beneficial in improving the quality of initial recommendations.

This paper introduces a versatile framework to capture specific events where new users engage with items, denoted as $(+1)$ or $(-1)$ for positive or negative interactions. This framework can be used in two cases: *i*) Predicting whether a user explicitly expresses an affinity for a product/item, signified by $(+1)$ for a positive review and $(-1)$ for a negative review, such as rating prediction. *ii*) Forecasting whether a user undertakes actions implicitly indicating a preference for an item, indicated by $(+1)$ when a user makes a purchase or $(-1)$ when no interaction with the item is recorded, such as news recommendation [Bae et al., 2023] and click-through-rate prediction [Zhang et al., 2022].

**Contributions.** To recommend items to new users, we propose a framework that estimates a region in the embedding space highly likely to contain the embedding of a new user. Our approach involves a preference elicitation process for selecting the best items to ask users to rate, considering that no initial information about the users is available. As a new user arrives, we prompt a static, short questionnaire with a carefully selected set of questions using a determinantal point process (DPP). The DPP ensures that the items listed in the questionnaire strike a balance between diversity and popularity (or quality), and the user's feedback on these items will serve as initial information about the users.

Our framework then focuses on constructing a *dynamic* questionnaire personalized to each user and sequentially updates our belief about the user's preference. By formulating and solving a minimization problem, we choose items that effectively narrow down the region in the embedding space where the new user's embedding is most likely to be found. This adaptive approach allows us to gather further information from users while limiting the questions to a relatively small number. Hence, our approach reduces the cognitive load on the user but can guarantee a good localization of the user's embedding simultaneously.

To enhance the practicality of our work, we introduce a user behavior model that incorporates a probabilistic assessment of whether a user has previous experience with an item. Users can provide feedback, either positively $(+1)$ or negatively $(-1)$, for items they have experienced. In practical cases where users have not experienced an item or choose to ignore the question, the feedback is NA, and this information is also taken into account to refine the selection of the items to query.

**Outline.** The subsequent sections unfold as follows: we

first introduce the problem settings and describe the user behavior model. Then, we present our solution package highlighting our Personalized Embedding Region Elicitation (PERE) method. The efficacy of our solution package is demonstrated through numerical results in the last section.

## 2 PROBLEM SETTINGS

The recommendation system has a list of $N$ items; each item is represented by a $d$-dimensional embedding vector $v_i \in \mathbb{R}^d$ for $i = 1, \ldots, N$. Additionally, we extract a popularity score for each item based on historical user-item interactions, represented by a normalized number $0 \leq w_i \leq 1$. The items are sorted in descending order of popularity, with $w_i \geq w_{i+1}$. The top-$P$ items are classified as *popular*, while the remaining $N - P$ items are considered non-popular. When a new user arrives without prior information, we aim to learn a suitable embedding vector for this user and then utilize this embedding vector for personalized item recommendations. Throughout, we rely on the assumption that the embeddings are of sufficient quality to enable distance-based recommendation methods such as k-nearest-neighbor to perform accurately. We make the following assumption:

**Assumption 1** (Embedding space). *The embeddings are normalized to a $d$-dimensional hypercube $\mathbb{H} = [0, 1]^d$. Moreover, the items' embeddings $v_i \in \mathbb{H}, \forall i = 1, \ldots, N$ do not change over time.*

This assumption imposes a bounded constraint on the embedding space, a common practice for machine learning algorithms. The invariance of item embeddings is also reasonable for most practical online platform systems where the items can be movies, books, or songs.

Our recommendation system employs a three-option feedback mechanism for user interactions. Whenever the user is presented with an item $i$ characterized by an embedding vector $v_i$, the user can rate the item using three options: $-1, +1$, or NA. A rating of $-1$ signifies a negative experience or dislike, while a rating of $+1$ indicates a positive experience or liking of the item. The user may also choose NA to express a lack of experience or refuse to disclose the preference. By employing this interactive scheme, we propose a two-phase preference elicitation process consisting of a burn-in phase and a sequential and adaptive question-answering (Q&A) phase. The elicitation process aims to extract the user's preferences and learn an appropriate embedding vector to represent the user in the common embedding space. We present the overall flow of our approach in Figure 1 and summarize the process in each phase as follows: when a new user arrives, we construct a burn-in questionnaire that consists of $K$ items to ask the user. The user rates $-1, +1$, or NA for each item in this list. By consolidating the responses from the user, we can divide the set $\mathcal{L}$ into three subsets: $\mathcal{L}^-$, $\mathcal{L}^+$, and $\mathcal{L}^{\text{NA}}$, that represent items disliked,

liked, and with no expressed opinion, respectively.[1]

Subsequently, we further facilitate the elicitation of user preferences through an adaptive Q&A process. Our system sequentially presents to the user $k$ new items in each round, and the user provides feedback ($-1$, $+1$, or NA) about the item to refine the identification of their embedding vector.

## 2.1 A MODEL OF THE USER'S BEHAVIOR

To create an interactive mechanism connecting the user and the recommender system, we need to build a behavior model for each user. Without any loss of generality, we assume that each user can be represented by an embedding vector $u_0 \in \mathbb{H}$. The true location of the vector $u_0$ remains elusive to the recommendation system, but it is *in*variant throughout the procedure of preference elicitation. The user can rate items positively ($+1$) or negatively ($-1$) only if they have prior experience with the item. For instance, in the context of Netflix, where items are movies, this translates to the user having watched certain movies. A key aspect of modeling user behavior revolves around the probability of experience. We make the following assumption on the probability that a user has experienced an item:

**Assumption 2** (Experience probability). *The probability that an user $u_0$ has experienced an item $v_i$ is given by*

$$p_{0i} \triangleq w_i \times \mathrm{sigmoid}\big(\frac{1}{c_{0i}} - \frac{\kappa_0}{\sqrt{d} - c_{0i}}\big), \qquad (1)$$

*where $c_{0i} = \|u_0 - v_i\|_2$ is the distance between the true user's and the item's embedding. Moreover, whether the user has prior experience with item $i$ is independent of whether the user has prior experience with any other item $j \neq i$.*

Assumption 2 proposes that the probability that a user has experienced an item depends on two main factors: the item popularity $w_i$ and the distance between the user's embedding and the item's embedding $c_{0i}$. According to Assumption 2, if two items, $i$ and $j$, have equal distances from the user's embedding, the item with higher popularity (indicated by a larger weight) will have a higher experience probability. The relationship between the experience probability and the distance between embeddings is complex. Notably, the Euclidean distance from $u_0$ to $v_i$ cannot exceed $\sqrt{d}$, where $d$ represents the dimension of the embedding hypercube $\mathbb{H}$. Additionally, the parameter $\kappa_0$ acts as a tolerance parameter known only to the user. When $c_{0i}$ approaches $0$, the sigmoid term tends to 1, and when $c_{0i}$ approaches $\sqrt{d}$, the sigmoid term tends to 0. To study the impact of parameter $\kappa$ on the probability that a user has experienced an item, we conduct an experiment in the supplementary.

---

[1]By construction, $\mathcal{L}^-$, $\mathcal{L}^+$, and $\mathcal{L}^{\mathrm{NA}}$ are exhaustive and mutually exclusive: their pairwise intersection is the empty set, and their union is $\mathcal{L}$.

Moreover, preference consistency is a fundamental question in the preference elicitation literature. Inconsistency in preference elicitation refers to situations when users provide conflicting or contradictory ratings or feedback for the same or similar items. For instance, if a user $u_0$ prefers item $v_i$ to $v_j$ but rates ($-1$) and ($+1$) for those two items, respectively. Thus, to ensure the consistency of our proposed method, we make the following assumption according to the preference consistency between a user and two items:

**Assumption 3** (Preference consistency). *Suppose that the true user's embedding is $u_0$. Given any two items $v_i$ and $v_j$ such that $\|u_0 - v_i\|_2 \leq \|u_0 - v_j\|_2$:*

*(i) If the user rates $v_j$ positively ($+1$), then the user can only rate $v_i$ either positively ($+1$) or with NA.*

*(ii) If the user rates $v_i$ negatively ($-1$), then the user can only rate $v_j$ either negatively ($-1$) or with NA.*

Assumption 3 ensures that the user's preference is consistent with the neighborhood structure of the embedding space. Inconsistency may arise if the user rates $v_i$ negatively and $v_j$ positively, even though $v_i$ is closer to the user's true embedding than $v_j$. This inconsistency is wholly eliminated under Assumption 3.

## 3 ADAPTIVE Q&A WITH PERSONALIZED EMBEDDING REGION ELICITATION

This section presents our proposed solution package comprising two distinct phases: a burn-in questionnaire and a sequential and adaptive Q&A process. Additionally, we provide a recommendation module based on the Chebyshev center of the region, which is designed specifically for the recommendation task. As there is no prior information about the user's preferences, we implement a burn-in phase using a determinantal point process (DPP) to generate a short, static questionnaire for each new user. The DPP balances two criteria: diversity and popularity.

The adaptive Q&A process facilitates the sequential elicitation of user preferences. We assume this phase lasts $T$ rounds; in each round, we select $m$ items to ask for feedback from the user. While the user's true embedding vector $u_0$ is not available to the system, we can characterize the plausible values of the user's embeddings from the user's feedback. By utilizing a set of positively rated items and negatively rated items, we can form pairwise preferences and effectively refine the plausible embedding region. Therefore, this iterative elicitation allows us to increase the accuracy of the preference approximation.

**Set of plausible embeddings.** We suppose the user has indicated a set of positively-rated items $\mathcal{L}^+$ and a set of negatively-rated items $\mathcal{L}^-$. The set of induced preferences

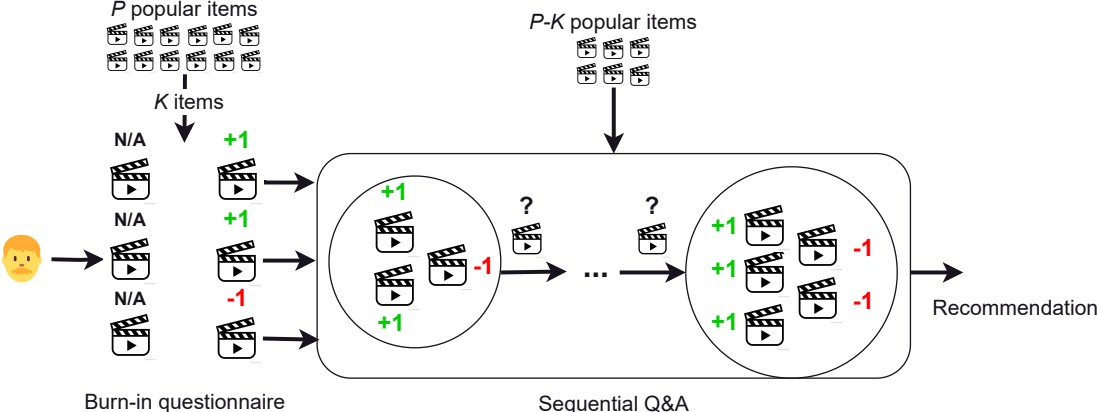

*P popular items*

*P-K popular items*

*K items*

N/A    +1

N/A    +1

N/A    -1

+1    ?    ?    +1    -1
-1         +1    -1
+1         +1    -1

Recommendation

Burn-in questionnaire

Sequential Q&A

Figure 1: When a new user arrives, we use a determinantal point process to query a diverse set of items from the *P popular* items list to construct the burn-in questionnaire. Subsequently, we use a sequential question-answering procedure to refine the embedding region of the user's preferences. The recommendation is made using the Chebyshev center of the embedding region, which is consistent with the user's stated preferences.

$\mathbb{P}$ is formed by picking any $i \in \mathcal{L}^+$ and any $j \in \mathcal{L}^-$, and appending the preference $v_i \succsim v_j$ to $\mathbb{P}$.[2] As a consequence, we have the following preference set

$$\mathbb{P} = \{v_i \succsim v_j : \forall v_i \in \mathcal{L}^+, \, \forall v_j \in \mathcal{L}^-\}. \qquad (2)$$

From any preference set $\mathbb{P}$, we can subsequently characterize a region $\mathcal{U}_\mathbb{P}$ that conforms with the user's preferences. For instance, if we pick any preference relation $v_i \succsim v_j$ in the preference set $\mathbb{P}$, Assumption 3 implies that the distance from the user's embedding $u_0$ to $v_i$ should be smaller than the distance to $v_j$. Because we are using Euclidean distance, this, in turn, implies that

$$\|v_i - u_0\|_2^2 \leq \|v_j - u_0\|_2^2.$$

By consolidating all preferences in the preference set $\mathbb{P}$, we expect the user's embedding to satisfy all of the below equations simultaneously. Thus, we have

$$\|v_i - u_0\|_2^2 \leq \|v_j - u_0\|_2^2 \quad \forall v_i \succsim v_j \in \mathbb{P}.$$

By expanding the norms, $u_0$ should satisfy

$$2u_0^\top (v_j - v_i) \leq \|v_j\|_2^2 - \|v_i\|_2^2 \quad \forall v_i \succsim v_j \in \mathbb{P}.$$

We denote $\mathcal{U}_\mathbb{P}$ as a set that contains all possible values of the embeddings that are consistent with the preference set $\mathbb{P}$, then we have

$$\mathcal{U}_\mathbb{P} = \{u \in \mathbb{H} : 2u^\top (v_j - v_i) \leq \|v_j\|_2^2 - \|v_i\|_2^2 \\ \forall v_i \succsim v_j \in \mathbb{P}\},$$

and under Assumption 3, we have $u_0 \in \mathcal{U}_\mathbb{P}$.

---

[2]For each user, we use $\succsim$ to denote a preference relation among items, that is, $\succsim$ denotes a complete and transitive order.

**Locating the Chebyshev center.** Now, we determine the Chebyshev center of the set $\mathcal{U}_\mathbb{P}$. The Chebyshev center refers to the center of a ball with the maximum radius and is enclosed within a bounded set with a non-empty interior. Consequently, the Chebyshev center of the confidence set $\mathcal{U}_\mathbb{P}$ is considered a safe point estimate for the true embedding $u_0$. Moreover, by identifying the Chebyshev center, we can find the most aggressive cut to the set $\mathcal{U}_\mathbb{P}$, thereby expediting the refinement of the plausible embedding region.

The Chebysev center $u_c^\star$ of the set $\mathcal{U}_\mathbb{P}$ and the radius $r^\star$ can be computed by solving the following problem

$$\max_{u_c \in \mathbb{H}, \, r \in \mathbb{R}_+} \left\{ r \; : \; \|u - u_c\|_2^2 \leq r^2 \; \forall u \in \mathcal{U}_\mathbb{P} \right\}.$$

For our problem, the Chebyshev center can be obtained by solving a linear program, resulting from the following theorem.

**Theorem 1** (Chebyshev center). *Suppose that $\mathcal{U}_\mathbb{P}$ has a non-empty interior. The Chebyshev center $u_c^\star$ of the set $\mathcal{U}_\mathbb{P}$ can be found by solving the following problem*

$$\begin{aligned} \max \quad & r \\ \text{s.t.} \quad & 2u_c^\top (v_j - v_i) + 2r\|v_j - v_i\|_2 \leq \|v_j\|_2^2 - \|v_i\|_2^2 \\ & \qquad\qquad\qquad\qquad \forall v_i \succsim v_j \in \mathbb{P} \\ & u_c \in \mathbb{H}, \, r \in \mathbb{R}_+. \end{aligned}$$

The proof of Theorem 1 follows from a duality argument and is relegated to the supplementary material.

**Next item to query.** At time $t+1$, we have already obtained user feedback on the list $\mathcal{L}_t$ of popular items, represented by the tuples $\mathcal{L}_t^+$, $\mathcal{L}_t^-$, and $\mathcal{L}_t^{\mathrm{NA}}$. The remaining popular items are denoted as $\mathcal{V}_t = \{v_i\}_{i=1,\dots,P} \backslash \mathcal{L}_t$. Then, for the next $T$ rounds, we select the next item $v_i$ from $\mathcal{V}_t$ and obtain the

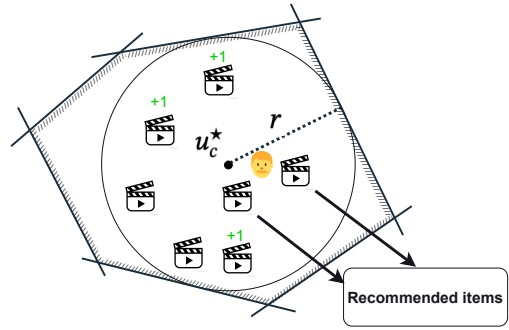

Figure 2: The hyperplanes $2u_c^\top(v_i - v_j) = \|v_i\|_2^2 - \|v_j\|_2^2$ for $i \succsim j \in \mathbb{P}$ are drawn as black lines, and they define the boundary of the set $\mathcal{U}_\mathbb{P}$. The ball centered at the Chebyshev center $u_c^\star$ with radius $r$ is the largest inscribed Euclidean ball of $\mathcal{U}_\mathbb{P}$. Our model recommends items based on the proximity to the Chebyshev center: here, two movies nearest to $u_c^\star$ are highlighted.

user's rating. The goal is twofold: if the user rates the newly presented item positively $(+1)$, we can leverage this positive experience along with the list of negatively-rated items $\mathcal{L}_t^-$ to generate new pairwise preferences. Conversely, if the user rates the item negatively $(-1)$, this information can be combined with the set $\mathcal{L}_t^+$ to create additional preferences. However, the feedback is uninformative if the user rates the new item as NA. Preferences involving two items $v_i$ and $v_j$ can be represented by a hyperplane equation

$$2u_c^\top(v_i - v_j) = \|v_i\|_2^2 - \|v_j\|_2^2.$$

A possible goal is to find the next item $v_i \in \mathcal{V}_t$ that minimizes the total weighted distance from the incumbent Chebyshev center to all constructed hyperplanes. To find the next item to ask, we need to consider the probability that the user has prior experience with the next item. The higher the probability, the more inclined the system should choose this item to obtain informative feedback (either a positive or a negative rating). Because the recommendation system does not know $u_0$ and $\kappa_0$, it does not know the true value of the probability that the user $u_0$ has prior experience with item $i$. Instead, the system will employ the following surrogate

$$\hat{p}_i = \widehat{\text{Prob}}(\text{user has experienced } v_i) \tag{3}$$

$$= w_i \times \text{sigmoid}\left(\frac{1}{\hat{c}_{0i}} - \frac{1}{\sqrt{d} - \hat{c}_{0i}}\right), \tag{4}$$

where $\hat{c}_{0i} = \|u_c^\star - v_i\|_2$. We can observe that this surrogate $\hat{p}_i$ does not depend on $\kappa_0$. Moreover, this surrogate probability is computed based on the distance from the item embedding to the incumbent Chebyshev center $u_c^\star$, but not to the true value of the user's embedding $u_0$. Our method is also robust to the misspecification of the functional form. The numerical section also shows that $\hat{p}_i$ can be calculated using the $\tanh$ function instead of the sigmoid function.

Conditioned that the user has prior experience with item $v_i$, there now exist three situations:

- Case 1: if the item $v_i$ satisfies

$$\|v_i - u_c^\star\|_2 \le \max_{v_j \in \mathcal{L}_t^+} \|v_j - u_c^\star\|_2,$$

then by Assumption 3(i) it is likely that the user will rate item $v_i$ positively $(+1)$ or NA. If we exercise optimism, we expect the user to rate positively $(+1)$. In this optimistic case, this positive rating from the user will lead to subsequently $|\mathcal{L}_t^-|$ new preferences of the form $v_i \succsim v_j$ for all $v_j \in \mathcal{L}_t^-$. Each pairwise preference is represented by a linear cut $2(v_j - v_i)^\top u \le \|v_j\|_2^2 - \|v_i\|_2^2$. The degree to which the above cut can effectively reduce the size of the set $\mathcal{U}_\mathbb{P}$ is quantified by the distance from the Chebyshev center $u_c^\star$ to the hyperplane $2u^\top(v_j - v_i) = \|v_j\|_2^2 - \|v_i\|_2^2$. An elementary calculation shows that this distance has an analytical form

$$\frac{|2(v_j - v_i)^\top u_c^\star + \|v_i\|_2^2 - \|v_j\|_2^2|}{\|2(v_j - v_i)\|_2}.$$

As a consequence, if we decide to sum up these distances, then the total distance from the Chebyshev center $u_c^\star$ to all hyperplanes generated by the positive feedback on item $v_i \in \mathcal{V}_t$ is

$$\sum_{v_j \in \mathcal{L}_t^-} \frac{|2(v_j - v_i)^\top u_c^\star + \|v_i\|_2^2 - \|v_j\|_2^2|}{\|2(v_j - v_i)\|_2} \triangleq q_i^+.$$

- Case 2: if the item $v_i$ satisfies

$$\|v_i - u_c^\star\|_2 \ge \min_{v_j \in \mathcal{L}_t^-} \|v_j - u_c^\star\|_2,$$

then by Assumption 3(ii), it is likely that the user will rate $v_i$ negatively $(-1)$ or NA. A parallel argument can quantify the total distance in this case:

$$\sum_{v_j \in \mathcal{L}_t^+} \frac{|2(v_i - v_j)^\top u_c^\star + \|v_j\|_2^2 - \|v_i\|_2^2|}{\|2(v_i - v_j)\|_2} \triangleq q_i^-.$$

- Case 3: if item $v_i$ does not satisfy the above conditions, then we have high uncertainty about the user's response for $v_i$. Nevertheless, if we opt for optimism, we can use the minimum of the two distances: $\min\{q_i^+, q_i^-\} \triangleq q_i^{\text{NA}}$.

Our goal is to choose the next items that maximize the probability of user experience for each chosen item while minimizing the distance from the resulting cut to the center in all three cases mentioned. Consequently, we can determine the next item to query by finding the equation below:

$$\min_{v_i \in \mathcal{V}_t} (1 - \hat{p}_i)\Big[q_i^+ \mathbb{I}^+(v_i) + q_i^- \mathbb{I}^-(v_i) +$$

$$q_i^{\text{NA}}(1 - \mathbb{I}^+(v_i))(1 - \mathbb{I}^-(v_i))\Big],$$

where $\mathbb{I}^+$ is the indicator function for Case 1 above:

$$\mathbb{I}^+(v_i) = \begin{cases} 1 & \text{if } \|v_i - u_c^\star\|_2 \leq \max_{v_j \in \mathcal{L}_t^+} \|v_j - u_c^\star\|_2, \\ 0 & \text{otherwise}, \end{cases}$$

and $\mathbb{I}^-$ is the indicator for Case 2:

$$\mathbb{I}^-(v_i) = \begin{cases} 1 & \text{if } \|v_i - u_c^\star\|_2 \geq \min_{v_j \in \mathcal{L}_t^-} \|v_j - u_c^\star\|_2, \\ 0 & \text{otherwise}. \end{cases}$$

Notice that these indicator functions depend on the current Chebyshev center $u_c^\star$ as well as the current set of positively-rated items $\mathcal{L}_t^+$ and negatively-rated items $\mathcal{L}_t^-$; however, these dependencies are omitted to avoid clutter.

To enhance understanding of our process for identifying $\mathcal{U}_\mathbb{P}$, we create and visualize a toy example comprising a single user and five items in the supplementary.

**Aggregating Chebyshev centers using a reweighting scheme.** Suppose that in the "burn-in" questionnaire, we have asked $K$ items. In sequential Q&A, suppose at round $t$, we obtain a Chebyshev center $u_c^t$ by solving the optimization problem in Theorem 1. Then, the aggregated center after the adaptive Q&A process (after $T$ rounds) can be computed using a reweighting scheme:

$$\bar{u}_c^\star = \sum_{t=0}^{T} \frac{K + t \times m}{K \times (T+1) + T \times (T+1) \times m/2} u_c^t. \quad (5)$$

The denominator is a normalizing constant so that the weights sum up to one.

**Item recommendation.** At any time, our system keeps track of three sets of items: $\mathcal{L}^+$, $\mathcal{L}^-$, and $\mathcal{L}^{\text{NA}}$. We generate all valid pairwise preferences by coupling items from the $\mathcal{L}^+$ and $\mathcal{L}^-$ sets. Each preference pair delineates a distinct cut in the embedding region, effectively narrowing down the area denoted by the set $\mathcal{U}_\mathbb{P}$ in the embedding space containing the new user embedding. To generate item recommendations, we calculate the Euclidean distance from all unqueried items to the aggregated center in (5) and recommend the top $k$ items nearest to this center.

# 4 NUMERICAL EXPERIMENTS

We conduct extensive experiments to study the efficacy of our proposed approach. We conduct two experiments to address the following research questions:

**RQ1:** Can our algorithm accurately approximate the embedding region that contains the new user's embeddings $u_0$ with minimal information?

**RQ2:** How does our item-selection mechanism proposed in Section 3 compare to baselines?

**RQ3:** How does our proposed method generalize to different types of embedding techniques and functional forms in estimating experience probability?

## 4.1 EXPERIMENT SETTINGS

**Datasets and User-Item Embedding.** In our experiments, we utilize two datasets: Gowalla and Amazon-Books. Both Amazon-Books and Gowalla datasets are standard benchmarks in the recommendation system literature. Many recent published papers use these two datasets, including Silva et al. [2023], Gong et al. [2023]. To process these datasets, we adhere to the pipeline in the LightGCN [He et al., 2020] and biVAE [Truong et al., 2021]. The embedding for users and items can be obtained using any collaborative filtering method, e.g., LightGCN [He et al., 2020] or biVAE [Truong et al., 2021]. Because of the lack of real data for the entire preference elicitation process in the cold-start recommendation problem, we consider the embedding produced by the collaborator filtering methods for new users (detailed in the next section) as the "true" embedding and conceal them from our algorithm. Our settings are still appropriate because we ensure that the recommender system does *not* have access to the ground truth embedding of the new user, and the system *only* has access to the user's behavior on the questionnaires. We design an experiment to see how well our algorithm approximates this "true" embedding for new users after a fixed number of questions. Note that the comparison of the collaborative filtering frameworks is beyond the scope of this paper. Additional information about these datasets and embedding generation can be found in the supplementary.

**New user's characteristics generation.** A generated user possesses four attributes: a user embedding denoted as $u_0$, an $N$-dimensional binary vector indicating whether the user has experienced each item, a list of liked items, and a list of disliked items. Generating a new user begins by obtaining the user embedding outlined in the above section. Then, based on the available data, we calculate the maximum likelihood estimate $\hat{\kappa}_{\text{MLE}}$, as detailed in supplementary. This estimation allows us to determine the user experience probability $p_i$ for each item as assumed in (1). To ascertain whether the new user has experienced a particular item, we generate a binary variable $z \in \{0, 1\}^N$ for each item using a Bernoulli distribution. If $z_i$ equals 0, the user has not experienced the item $i$ (NA). Conversely, if $z_i$ equals 1, the user has previously experienced the item $i$. Additionally, we retrieve the top $k$ items closest to $u_0$ and append them to the list of liked items. Let $N_e$ be the number of items experienced by the user. If the user has only experienced $N_e$ items, the remaining $N_e - k$ items are considered to be disliked items for this user.

**Setup.** We employ DPP to curate a diverse set of items ($K = 50$ items) for inquiries directed at newly registered users. Supporting evidence demonstrating the superior efficiency of DPP compared to competing methods such as greedy and random generation is included in the supplementary material. In the sequential Q&A phase, we present the user

with $T = 5$ questionnaires; each contains $m = 10$ items.

**Baselines.** We compare our proposed method PERE (Personalized Embedding Region Elicitation) against six baselines: DPP, Conditional DPP (c-DPP), RMV [Fonarev et al., 2016], DPE [Parapar and Radlinski, 2021], PEO [Sepliarskaia et al., 2018] and DRE Kweon et al. [2020]. c-DPP is a modified version of DPP that selects $K$ items from the remaining un-queried items. We note that for a fair comparison, we must compare our proposed method against other cold-start recommendation methods with preference elicitation. Our chosen baselines are the most recent methods in that line of research work.

**Performance Metric.** To assess different approaches, we employ several metrics. These include NDCG@$k$ (Normalized Discounted Cumulative Gain), which evaluates relevance and ranking simultaneously; MAP (Mean Average Precision), providing an aggregate measure of precision; and MRR (Mean Reciprocal Rank), indicating promptness in presenting relevant items. These metrics collectively estimate the recommendation system's accuracy, relevance, ranking quality, and user satisfaction.

Table 1: Benchmark of performance metrics on Amazon-Books (user and item embeddings produced by biVAE). Larger values are better. The best performance for any fixed number of questions is highlighted in bold. Sequential Setting contains $50 + 10 + 10 + 10 + 10 + 10$ items.

| Elicited items | Method | NDCG@10 | MAP | MRR |
|---|---|---|---|---|
| 100 items | RMV | 0.1898 | 0.1349 | 0.176 |
| 100 items | DPE | 0.2096 | 0.1546 | 0.1923 |
| 100 items | PEO | 0.2218 | 0.1184 | 0.1552 |
| 100 items | DRE | 0.1133 | 0.0702 | 0.0841 |
| Sequential | c-DPP | 0.2901 | 0.2229 | 0.2599 |
| Sequential | PERE | **0.2918** | **0.2265** | **0.2695** |

Table 2: Benchmark of performance metrics on Amazon-Books (user and item embeddings produced by LightGCN). Larger values are better. The best performance for any fixed number of questions is highlighted in bold. Sequential Setting contains $50 + 10 + 10 + 10 + 10 + 10$ items.

| Elicited items | Method | NDCG@10 | MAP | MRR |
|---|---|---|---|---|
| 100 items | RMV | 0.2372 | 0.1946 | 0.2175 |
| 100 items | DPE | 0.2443 | 0.2007 | 0.2234 |
| 100 items | PEO | 0.3108 | 0.2597 | 0.2914 |
| 100 items | DRE | 0.0382 | 0.0214 | 0.0431 |
| Sequential | c-DPP | 0.3575 | 0.2842 | 0.3151 |
| Sequential | PERE | **0.3616** | **0.2930** | **0.3235** |

Table 3: Benchmark of performance metrics on Gowalla (user and item embeddings produced by LightGCN). Larger values are better. The best performance for any fixed number of questions is highlighted in bold. Sequential Setting contains $50 + 10 + 10 + 10 + 10 + 10$ items.

| Elicited items | Method | NDCG@10 | MAP | MRR |
|---|---|---|---|---|
| 100 items | RMV | 0.0711 | 0.0317 | 0.0598 |
| 100 items | DPE | 0.0846 | 0.0587 | 0.0687 |
| 100 items | PEO | 0.1307 | 0.0952 | 0.1141 |
| 100 items | DRE | 0.1037 | 0.0499 | 0.0768 |
| Sequential | c-DPP | 0.1764 | 0.1287 | 0.1461 |
| Sequential | PERE | **0.1806** | **0.1309** | **0.1518** |

## 4.2 NUMERICAL RESULTS AND DISCUSSION

The numerical results on different datasets and embeddings are summarised into Tables 1, 2, and 3. Due to space limitations, we report experimental results with more performance metrics in the supplementary.

**Recommendation quality.** The results indicate that our method is the most effective method for constructing a personalized series of follow-up questions for new users. Building upon the success of DPP, the best-performing method in the "burn-in" phase, PERE exhibits the most significant improvement in quality after 50 items have been asked. This is significant due to the small experience probability of items, as defined in (1). Despite the limited information users provide, our framework successfully enhances the quality of recommendations based on this minimal input. Addressing any potential question about this enhancement stemming solely from the sequential nature of our framework, we also conduct comparisons with other sequential methodologies, such as bandit (DPE), conditional DPP, and active learning (PEO). The results conclusively demonstrate that even when evaluated among sequential methods, PERE remains the top performer.

**Generalizability.** Comparing Tables 1 and 2, we observe that our method PERE efficiently generalizes to multiple types of embedding generation techniques, in our case, LightGCN (trained with implicit user response) and biVAE (trained with explicit user response).

**Robustness with functional misspecification.** To evaluate the robustness against misspecification of the functional form in estimating experience probability, we devise an experiment utilizing both the sigmoid and tanh functions in equation (3). Table 4 highlights that our method consistently upholds recommendation quality despite replacing the sigmoid with the tanh function.

**Real user experiments.** We design additional offline experiments using real user data accumulated in datasets such as MovieLens 10M, MovieLens 20M, and Amazon Books.

Table 4: Comparing sigmoid and tanh in estimating equation (3): our method is robust to the functional misspecification.

| Datasets | Method | HR@1 | AUC@10 | NDCG@10 | NDCG@30 | MAP | MRR |
|---|---|---|---|---|---|---|---|
| Gowalla | PERE (tanh) | 0.0767 | 0.1872 | 0.1804 | 0.2015 | 0.1303 | 0.1516 |
| | PERE (sigmoid) | 0.0767 | 0.1879 | 0.1806 | 0.2017 | 0.1309 | 0.1518 |
| Amazon-Books (LightGCN) | PERE (tanh) | 0.2133 | 0.3512 | 0.3629 | 0.3865 | 0.2961 | 0.3245 |
| | PERE (sigmoid) | 0.21 | 0.3539 | 0.3616 | 0.3872 | 0.293 | 0.3235 |
| Amazon-Books (biVAE) | PERE (tanh) | 0.1833 | 0.2587 | 0.2918 | 0.3307 | 0.2274 | 0.2693 |
| | PERE (sigmoid) | 0.1833 | 0.2588 | 0.2918 | 0.3314 | 0.2265 | 0.2695 |

We focused on the Amazon Books, MovieLens 10M, and MovieLens 20M datasets because they are widely used in the recommender systems literature and provide a diverse set of user-item interactions across different domains. These datasets are comparable in scale and complexity to those used in the related work section, making them suitable for evaluating the generalizability of our framework.

The items users prefer are based on their actual ratings from the datasets rather than being generated from the embeddings. We followed the data processing approach used in the Deep rating elicitation [Kweon et al., 2020], which is also one of our baselines. Specifically, we filtered out users who rated over $40$ items and converted implicit ratings $(1-5)$ to explicit ratings (0 and 1) as follows: ratings of $4$ and $5$ are considered as liked items, while ratings of $0$, $1$, and $3$ are considered as disliked items. Table 5 demonstrates that our method outperforms baselines in all three datasets.

**Inconsistent preference.** We conduct two additional experiments to evaluate our method's performance under inconsistent user preferences:

- **Experiment 1:** We introduce a probability $\tau$ that a user's response to an experienced item will be flipped. When $\tau = 0$, there is no inconsistency, and when $\tau = 1$, responses are always inconsistent. We plot the performance gain in NDCG@50 against the number of displayed items for different values of $\tau$. Figure 3 shows that as $\tau$ increases, the performance gain decreases but remains positive, demonstrating that our method still provides benefits despite inconsistencies in user responses.

- **Experiment 2:** We compare our method against DPE and RMV in the presence of inconsistency. Table 6 shows that our method maintains its advantage over the baselines even with inconsistent user preferences.

**Questionnaire size analysis.** To be user-friendly, the questionnaire size should be small to avoid stressing the user's cognitive load. We find that the number of items at each round does not significantly affect the quality of the method. What is more interesting to track is the quality improvement over a long history as the *total* number of questions increases. Therefore, we conduct an additional experiment

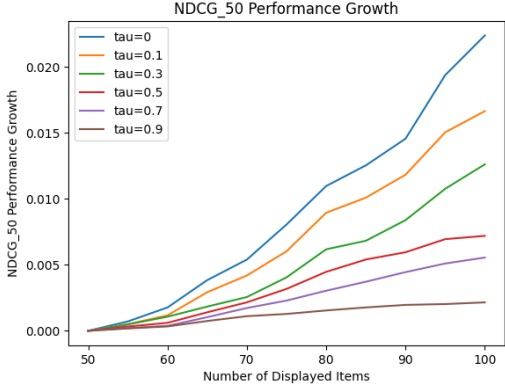

Figure 3: As the value of $\kappa_0$ increases, NDCG@50 increases under inconsistent preference setting.

to study the impact of the total number of questions on the performance metrics NDCG@10 and MRR. Figure 4 shows that our method outperforms RMV, DPE, PEO, and DRE in all datasets. Additionally, our method outperforms c-DPP when eliciting $K = 100$ items in total.

**Runtime comparison.** We report the run-time experiments on our largest dataset, Amazon-Books: The average run time per round is approximately 0.27 seconds. We believe this runtime is reasonable for real-time systems if we optimize the hardware-software for deployment.

## 5 RELATED WORKS

**Cold-start recommendation.** The cold-start problem presents a significant challenge within recommender systems. This challenge emerges from the sparsity of information necessary to personalize recommendations for users effectively. In most cases, users and items have limited or no interactions. Several approaches have been proposed to tackle the cold-start problem for recommender systems [Rajapakse and Leith, 2022, Guo et al., 2020, Camacho and Alves-Souza, 2018]. A possible solution for tackling the cold-start problem is to employ collaborative filtering techniques [Natarajan et al., 2020, Wei et al., 2020, Anwar et al., 2022]. For instance, Son and Kim [2017] introduced a hy-

Table 5: Real user experiments on three datasets.

| Datasets | Methods | NDCG@10 | MAP | MRR |
|---|---|---|---|---|
| MoviesLens-10M | c-DPP | 0.802 | 0.591 | 0.772 |
| | DPE | 0.674 | 0.439 | 0.607 |
| | RMV | 0.492 | 0.281 | 0.437 |
| | PEO | 0.667 | 0.429 | 0.69 |
| | DRE | 0.337 | 0.164 | 0.271 |
| | PERE | **0.812** | **0.603** | **0.784** |
| MoviesLens-20M | c-DPP | 0.628 | 0.499 | 0.689 |
| | DPE | 0.634 | 0.428 | 0.578 |
| | RMV | 0.144 | 0.088 | 0.133 |
| | PEO | 0.639 | 0.394 | 0.592 |
| | DRE | 0.435 | 0.227 | 0.364 |
| | PERE | **0.734** | **0.505** | **0.696** |
| Amazon-Books | c-DPP | 0.127 | 0.101 | 0.108 |
| | DPE | 0.082 | 0.078 | 0.084 |
| | RMV | 0.044 | 0.041 | 0.047 |
| | PEO | 0.099 | 0.084 | 0.105 |
| | DRE | 0.029 | 0.025 | 0.027 |
| | PERE | **0.132** | **0.106** | **0.125** |

Table 6: Comparison against DPE and RMV under inconsistent preference setting.

| Datasets | Methods | HR@5 | NDCG@10 | MRR |
|---|---|---|---|---|
| Amazon-Books ($\tau = 0.1$) | DPE | 0.305 | 0.285 | 0.293 |
| | RMV | 0.205 | 0.188 | 0.179 |
| | PERE | **0.365** | **0.303** | **0.329** |
| Amazon-Books ($\tau = 0.5$) | DPE | 0.310 | 0.288 | 0.297 |
| | RMV | 0.205 | 0.188 | 0.179 |
| | PERE | **0.360** | **0.329** | **0.328** |

brid approach that combines collaborative filtering with content-based methods to mitigate the cold-start problem.

Deep learning techniques have been employed to learn representations or embeddings that capture the latent features of users and items to handle the cold-start problem [Tao et al., 2022, Raziperchikolaei et al., 2021, Chu et al., 2023, Yu et al., 2021, Zheng et al., 2021]. Recently, graph-based recommendation techniques have become effective approaches for learning user and item representations [Ying et al., 2018, Salha-Galvan et al., 2021]. These methods leverage the user-item interaction graphs to infer user preferences. For example, Ying et al. [2018] develops a graph autoencoder framework to learn the node representation. This approach empirically shows competitive performance under real-world scenarios.

**Rating elicitation.** Rating elicitation plays a crucial role in recommender systems, as it involves gathering explicit user feedback to understand their preferences. Rating elicitation refers to a Q&A process employed by a system to request new users to rate a set of items. This process aims to infer the users' preferences and enhance the quality of the recommendations. The primary challenge in rating elicitation lies

in selecting the seed items that can effectively capture the new users' preferences.

One of the first approaches to solving rating elicitation is Active Collaborative Filtering (CF). Most Active CF methods ask users to rate the set of items that maximize the Expected Value of Information Boutilier et al. [2002], Harpale and Yang [2008], information gain Canal et al. [2019], Rashid et al. [2002], Houlsby et al. [2014], influence criterion Rubens and Sugiyama [2007] or minimize the estimated model Entropy Jin and Si [2004], Houlsby et al. [2012]. However, those methods rely on the current estimated model, which is obtained via a few user's warm-start ratings instead of a completely cold-start user setting.

Notably, region refining methods closely resemble our work. For example, Iyengar et al. [2001] proposes Q-Eval, a preference elicitation method that iteratively refines a permissible region over the weights of multiple item attributes. Another method Toubia et al. [2004] involves selecting questions by adding cuts to narrow down the feasible region defined by a polyhedron. However, these methods consider a lower dimensional space compared to our work, which involves a higher-dimensional embedding space. Additionally, our

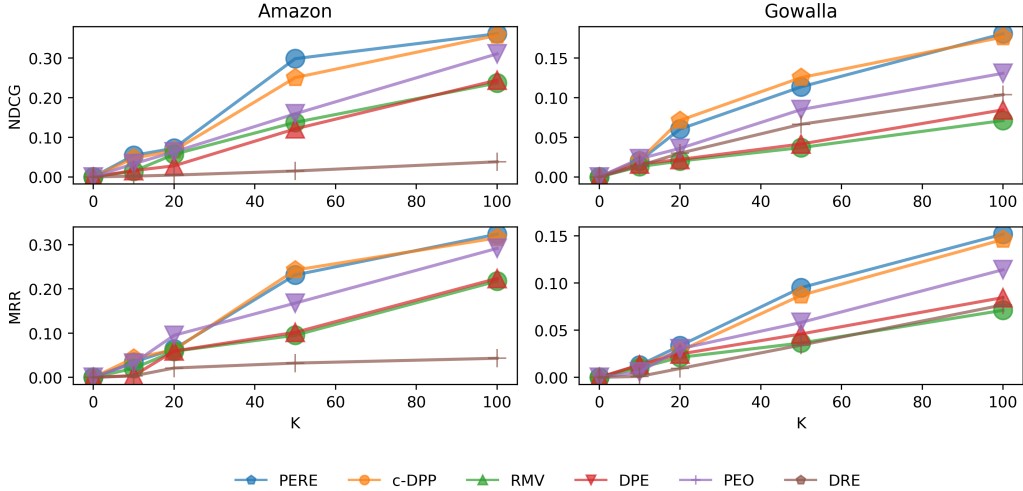

Figure 4: Performance improvements with the dynamic questionnaire size on Amazon-Books and Gowalla datasets.

question selection criteria are more complex, combining diversity and information gain maximization.

Recently, rating elicitation has emerged as a powerful method [Gope and Jain, 2017, Pu et al., 2012] to tackle the cold-start problem in recommender systems. For instance, Kalloori et al. [2018] proposed an active learning method for pairwise items and a personalized ranking algorithm to increase user satisfaction. Parapar and Radlinski [2021] employed multi-armed bandits, a well-established exploration-exploitation framework from reinforcement learning, to diversify the preferences elicited by the recommendation model. However, in real-world settings, these approaches rely on a fundamental assumption that users will consistently provide feedback, regardless of whether they have experienced the item or not. Nonetheless, this assumption may not hold true in practice. In this work, we address this problem by proposing a novel behavior model for the user and a preference elicitation process that directly takes the experience probability into consideration.

## 6 CONCLUSION

In this paper, we have addressed the problem of cold-start recommendation by proposing a personalized elicitation scheme consisting of two phases. After a short "burn-in" phase, we employ an adaptive preference approach where users are sequentially prompted to rate items that refine their preferences and user representation. Throughout the process, the system represents the user's preferences as a region estimate rather than a single point, capturing the uncertainty in their preferences. The value of information gained from user ratings is quantified by considering the distance from the region center that confidently contains the true embedding value. Recommendations are generated by considering the user's preferences region. We have demonstrated the

efficiency of each subproblem in the elicitation scheme and conducted empirical evaluations on prominent datasets to showcase the effectiveness of our proposed method compared to existing rating-elicitation approaches.

**Acknowledgments.** Viet Anh Nguyen gratefully acknowledges the generous support from the CUHK's Improvement on Competitiveness in Hiring New Faculties Funding Scheme and the CUHK's Direct Grant Project Number 4055191.

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

# Supplementary Material for Paper: Cold-start Recommendation by Personalized Embedding Region Elicitation

**Hieu Trung Nguyen**[1,3]      **Duy Nguyen**[1]      **Khoa Doan**[2]      **Viet Anh Nguyen**[3]

[1]VinAI Research
[2]College of Engineering & Computer Science, VinUniversity
[3]The Chinese University of Hong Kong

## A   PROOF OF THEOREM 1

We here provide the proof of Theorem 1 that are omitted in the main text.

*Proof.* The optimization problem to find the Chebyshev center and its radius can be rewritten as

$$
\begin{aligned}
\max \quad & r \\
\text{s.t.} \quad & 2(u_c + \delta)^\top (v_j - v_i) \leq \|v_j\|_2^2 - \|v_i\|_2^2 \\
& \qquad \forall \delta \in \mathcal{B}_r, \ \forall v_i \succsim v_j \in \mathbb{P} \\
& u_c \in \mathbb{H}, \ r \in \mathbb{R}_+,
\end{aligned}
$$

where $\mathcal{B}_r = \{\delta \in \mathbb{R}^d : \|\delta\|_2 \leq r\}$ is a $d$-dimensional Euclidean ball of radius $r$. Pick any preference $v_i \succsim v_j \in \mathbb{P}$, the semi-infinite constraint

$$
2(u_c + \delta)^\top (v_j - v_i) \leq \|v_j\|_2^2 - \|v_i\|_2^2 \ \forall \delta \in \mathcal{B}_r
$$

is equivalent to the robust constraint

$$
2u_c^\top (v_j - v_i) + 2 \sup_{\|\delta\|_2 \leq r} \delta^\top (v_j - v_i) \leq \|v_j\|_2^2 - \|v_i\|_2^2.
$$

Because the Euclidean norm is a self-dual norm, we have

$$
\sup_{\|\delta\|_2 \leq r} \delta^\top (v_j - v_i) = r\|v_j - v_i\|_2.
$$

Substituting the above relationship to the optimization problem completes the proof. □

## B   FURTHER EXPLANATIONS ABOUT SETTINGS AND REGION ELICITATION

In Assumption 2 , the probability that an user $u_0$ has experienced an item $v_i$ is given by

$$
p_{0i} \triangleq w_i \times \text{sigmoid}\Big(\frac{1}{c_{0i}} - \frac{\kappa_0}{\sqrt{d} - c_{0i}}\Big),
$$

where $c_{0i} = \|u_0 - v_i\|_2$ is the distance between the true user's and the item's embedding. In Figure 5, we visualize the dependence of $p_{0i}$ on the parameter $\kappa_0$. For a fixed value of the distance $c_{0i}$, the experience probability $p_{0i}$ decreases monotonically in $\kappa_0$.

Next, in a toy 2D example, we visualize the region $\mathcal{U}_\mathbb{P}$ in Figure 6. Initially, a new user (red star) came into our system, but we are unaware of its true embedding location. After two steps of elicitation, it is evident that the Chebyshev center moves progressively closer to the 'True User' embedding, underscoring the success of our proposed method in predicting user embeddings.

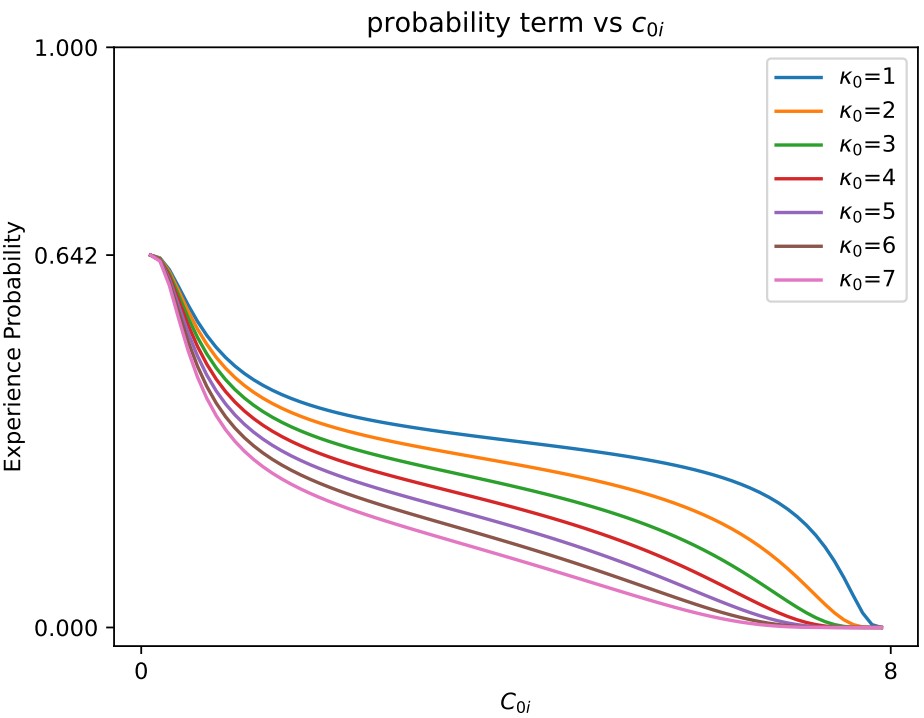

Figure 5: As the value of $\kappa_0$ increases, the probability that the user has prior experience (see Assumption 2) with an item is dampened. Plot with $d = 64$ and the maximal value of $c_{0i}$ is $\sqrt{d} = 8$.

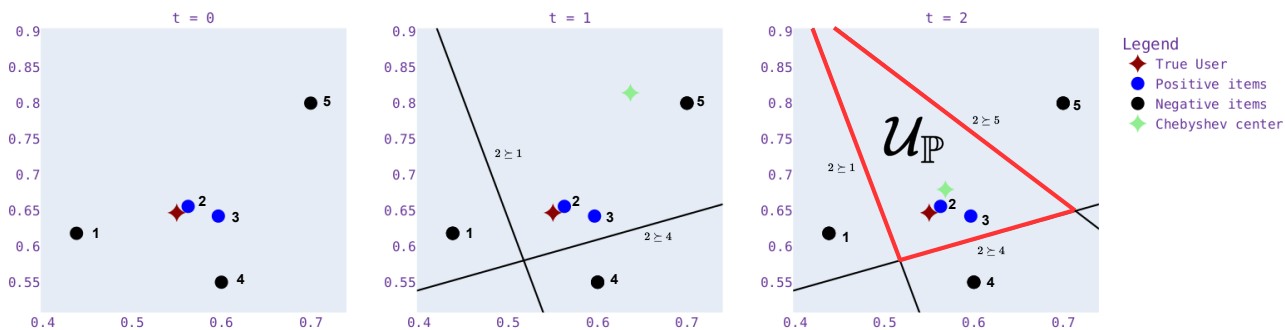

Figure 6: Illustration of our method in 2D toy example: Recall that a cut in the embedding space is created by pairing a positive item with a negative item. At time $t = 0$, when no questions have been asked, there are no cuts in the embedding space. Moving to time $t = 1$, we asked the user to elicit items 1, 2, and 4, and the user-specified 'dislike,' 'like,' and 'dislike' for each respective item. This introduces two cuts in the space, and the initial Chebyshev center is calculated. Progressing to time $t = 3$, we ask the user to elicit item 5 and determine it to be a disliked item. As a result, a final cut is constructed by pairing item 2 with item 5. This process concludes with the finalization of region $\mathcal{U}_{\mathbb{P}}$

## C  COLD-STARTING QUERY LIST VIA DETERMINANTAL POINT PROCESSES

The main task of the "burn-in" Phase is to create a list, denoted as $\mathcal{L}$, comprising $K$ popular items for querying the new user. If a user has no previous experience with an item $v_i$, they will indicate NA for that particular item. This NA response is uninformative because item $v_i$ does not lead to any pair of preferences being added to the preference list $\mathbb{P}$ as by the rule of preference construction. Therefore, when constructing the cold-start item list $\mathcal{L}$, it is important to consider the probability that a user has prior experience with the items. By Assumption 2, this probability is affected by two elements: the popularity of the item and the distance from the true user embedding $u_0$ to the item embedding $v_i$.

Since we do not know the user embedding $u_0$, but we have information about the popularity of the items, we thus leverage this popularity information in the construction of $\mathcal{L}$. This line of argument also justifies the construction of the list $\mathcal{L}$ that contains only the most popular items from the list of all possible items. To find this list $\mathcal{L}$, we can use a simple weighted $K$-medoids method: given a list of $N$ items; the weighted $K$-medoids return a subset of $K$ items to be used as cluster centers. The weighted $K$-medoids problem aims to minimize the total weighted squared Euclidean distance from the item embeddings to the nearest centers.

We present in this section a determinantal point process (DPP) to construct the item list $\mathcal{L}$. We aim to find a set of items that can balance the diversity and popularity of items oblivious to the user's true embedding. DPPs are elegant probabilistic models of global, negative correlations, and they admit efficient algorithms for sampling, marginalization, conditioning, and other inference tasks [Kulesza et al., 2012]. DPPs have been applied in various machine learning tasks, including document summarization [Perez-Beltrachini and Lapata, 2021] and image search [Chao et al., 2015]. We rely on the following $L$-ensemble definition of DPP.

**Definition 1** ($L$-ensemble DPP). *Given a positive semidefinite $P$-by-$P$ matrix $L \in \mathbb{S}_+^P$, an $L$-ensemble DPP is a distribution over all $2^P$ index subsets $J \subseteq \{1, \ldots, P\}$ such that*

$$\mathrm{Prob}(J) = \det(L_J)/\det(I + L),$$

*where $L_J$ denotes the $|J|$-by-$|J|$ submatrix of $L$ with rows and columns indexed by $J$.*

We design the matrix $L$ that can balance the diversity and popularity of items. We compose $L$ as the sum of a similarity matrix $S$ and a popularity matrix $D$ among items:

$$L = S + D, \quad \text{where} \quad D = \mathrm{diag}(w_i).$$

The matrix $D$ is diagonal, and its diagonal elements capture the popularity of the items. A possible choice for the similarity matrix $S$ is $S = V^\top V \in \mathbb{S}_+^P$ where $V$ is the embedding matrix of the popular items. Because both $S$ and $D$ are positive semidefinite, the ensemble matrix $L$ is also positive semidefinite.

We then find the combination of top-$K$ items that fit with the construction of the cold-start querying list by solving the following problem

$$\max \left\{ \det(L_z) \ : \ z \in \{0,1\}^P, \ \|z\|_0 = K \right\}, \tag{6}$$

where $L_z$ is a submatrix of $L$ restricted to rows and columns indexed by the one-components of $z$. It is well-known that the solution to problem (6) coincides with the MAP estimate of the DPP with a cardinality constraint [Kulesza et al., 2012]. Further, it is crucial to highlight that problem (6) is a submodular maximization problem since the log-probability function $\log \det(L_z)$ is a submodular function [Gillenwater et al., 2012]. Further, this problem is well-known to be NP-hard [Kulesza et al., 2012], and thus it is notoriously challenging to solve (6) to optimality. Chen et al. [2018] provides a greedy algorithm for the MAP estimation problem. The aforementioned greedy algorithm has been proven to achieve an approximation ratio of $\mathcal{O}(\frac{1}{k!})$ [Civril and Magdon-Ismail, 2009] and incur a computational complexity of $\mathcal{O}(K^2 P)$. Moreover, to improve the solution quality, we introduce a 2-neighborhood local search strategy. This method involves an iterative process of exchanging one element from the current set with one element from the complementary set, continuing until no additional improvement can be achieved.

# D   MAXIMUM LIKELIHOOD ESTIMATION OF THE TOLERANCE PARAMETER

We provide the maximum likelihood estimation for the parameters $\kappa$. Without any loss of generality, we consider a training dataset consisting of $N$ items and $M$ users; the user embeddings $u_m$, and the item embeddings $v_i$ are given. The interactions between the users and the items are presented by a binary-valued data matrix $E \in \{0,1\}^{M \times N}$ with each $E_{mi}$ admits values

$$E_{mi} = \begin{cases} 1 & \text{if user } m \text{ has an experience with item } i, \\ 0 & \text{otherwise.} \end{cases}$$

Suppose that there exists a global constant $\kappa \in \mathbb{R}_+$ such that $E_{mi}$ follows a Bernoulli random variable with

$$\mathrm{Prob}(E_{mi} = 1) = w_i \times \mathrm{sigmoid}\big(\frac{1}{c_{mi}} - \frac{\kappa}{\sqrt{d} - c_{mi}}\big),$$

where $c_{mi}$ is the embedding distance between the user the the item $c_{mi} = \|u_m - v_i\|_2$. Given the data matrix $E$ and suppose that the elements $E_{mi}$ are jointly independent, the likelihood is

$$L(\kappa|E) = \prod_{m=1}^{M} \prod_{i=1}^{N} (p_{mi}(\kappa))^{E_{mi}} (1 - p_{mi}(\kappa))^{1-E_{mi}},$$

where $p_{mi}(\kappa)$ is

$$p_{mi}(\kappa) = \frac{w_i}{1 + \exp\left(\frac{\kappa}{\sqrt{d}-c_{mi}} - \frac{1}{c_{mi}}\right)}.$$

The estimate $\hat{\kappa}_{\mathrm{MLE}}$ minimizes the negative log-likelihood:

$$\min_{\kappa \geq 0} \sum_{m=1}^{M} \sum_{i=1}^{N} \log\left(1 + \exp\left(\frac{\kappa}{\sqrt{d} - c_{mi}} - \frac{1}{c_{mi}}\right)\right)$$
$$- \sum_{m=1}^{M} \sum_{i=1}^{N} (1 - E_{mi}) \log\left(1 + \exp\left(\frac{\kappa}{\sqrt{d} - c_{mi}} - \frac{1}{c_{mi}}\right) - w_i\right),$$

which can be found by standard gradient descent algorithms.

# E    QUESTIONNAIRE DESIGN

Inspired by the structure of the Netflix questionnaire [Kweon et al., 2020], we devise our questionnaire methodology to capture a comprehensive set of preference pairs while minimizing user effort. Users are provided the option to skip specifying preferences, streamlining the process. In our questionnaire, users are presented with a product display, and while scrolling through, they only need to indicate 'like' or 'dislike' for products they are familiar with. An illustration of the questionnaire is provided in Figure 7. In practice, although our experimental design prompts new users to specify preferences for 100 items, our algorithm performs effectively even when utilizing an average of around 15% of user responses, evident by the user response ratio in Table 7.

Figure 7: Illustration of our questionnaire: Taking inspiration from the Netflix questionnaire as outlined in Kweon et al. [2020], we structure each questionnaire as depicted above. Upon a new user entering our system, we prompt them to indicate their preferences for a set of items. Users can specify 'like' $(+1)$, 'dislike' $(-1)$, or choose to skip the item (NA).

Table 7: Number of items responded to by users using the PERE method. The response ratio is calculated over 100 queried items.

| Dataset | Liked items | Disliked items | Response ratio |
|---------|-------------|----------------|----------------|
| Amazon-Books | 1.6333 | 14.7733 | 16.4066 % |
| Gowalla | 1.0000 | 11.2623 | 12.2623 % |

# F  ADDITIONAL NUMERICAL RESULTS

## F.1  STATISTICAL TEST

For each user, we compute the recommendation metrics for our methods and baselines. We propose to test the hypotheses:

- Null hypothesis: PERE's NDCG@10 (or MAP, MRR) equals the competing method's NDCG@10 (or MAP, MRR)

- Alternative hypothesis: PERE's NDCG@10 (or MAP, MRR) is larger than the competing method's NDCG@10 (or MAP, MRR).

In order to test the above hypothesis, we use a one-sided Wilcoxon signed-rank test to compare the paired metric values. Suppose we choose the significant level at 0.05. Table 8 indicates that PERE significantly outperforms RMV, DPE, PEO, and DRE across all performance metrics. PERE outperforms c-DPP in almost all metrics except for the NDCG@10 and MAP in the Gowalla dataset. However, this does not imply that c-DPP's NDCG@10 and MAP are higher than PERE in the Gowalla dataset.

Table 8: Statistical tests of 3 recommendation metrics across Amazon-Books and Gowalla datasets.

| Metrics | Datasets | PERE vs. RMV | PERE vs. DPE | PERE vs. PEO | PERE vs. DRE | PERE vs. c-DPP |
|---------|----------|--------------|--------------|--------------|--------------|----------------|
| NDCG@10 | Amazon-Books | $3e-12$ | $4e-11$ | $5e-3$ | $2e-26$ | 0.014 |
|  | Gowalla | $1e-14$ | $3e-14$ | $2e-3$ | $5e-10$ | 0.198 |
| MAP | Amazon-Books | $2e-6$ | $4e-5$ | $1e-3$ | $1e-33$ | 0.026 |
|  | Gowalla | $3e-12$ | $3e-9$ | $9e-5$ | $3e-7$ | 0.087 |
| MRR | Amazon-Books | $4e-11$ | $4e-11$ | $9e-3$ | $2e-31$ | 0.016 |
|  | Gowalla | $2e-9$ | $4e-7$ | $7e-4$ | $6e-6$ | 0.039 |

## F.2  BURN-IN PHASE COMPARISON

We use LightGCN / BiVAE for the burn-in phase to generate item embedding and conduct experiments on Gowalla and Amazon-Books datasets. We employ two widely recognized and straightforward baseline methods: RMV [Fonarev et al., 2016] and $K$-Medoids [Liu et al., 2011]: RMV optimizes the volume of a rectangle matrix by selecting diverse yet orthogonal seed items in the embedding space. On the other hand, the $K$-Medoids algorithm, previously employed in a study [Liu et al., 2011], identifies representative items through cluster centroids. We slightly modify the $K$-medoids algorithm by considering only the items belonging to the popular items as potential centroids. Note that sequential-based preference elicitation methods, such as DPE [Parapar and Radlinski, 2021] or conditional DPP, are not applicable during the 'burn-in' phase. In this phase, we aim to create a standardized questionnaire for all new users entering our system. Sequential-based methods, in contrast, involve asking new questions based on the responses of previous users.

Results for the burn-in phase are summarised in Table 9. The results demonstrate that DPP (Determinantal Point Process) is the best approach for selecting initial items for the initial queries. DPP significantly outperforms baseline methods regarding performance metrics in all two datasets. The success of DPP can be attributed to its ability to effectively select a diverse set of items while considering the popularity score of each item. This combination allows DPP to balance diversity and relevance, resulting in superior performance compared to the baseline methods.

Moreover, to show the effectiveness of our proposed sequential elicitation in Section 3, we conduct an additional experiment that compares PERE, which uses a static 50-item questionnaire in the beginning, and a series of 5 dynamic 5-item questionnaires after that, with a baseline where only a burn-in questionnaire using DPP is utilized to create a static 100-item

Table 9: Benchmark of performance metrics on Gowalla and Amazon-Books. Larger values are better. The best performance for any fixed number of questions is highlighted in bold. The number of items, in this case, is $K = 50$ for all methods.

| Dataset | Method | HR@5 | HR@10 | AUC@5 | AUC@10 | NDCG@10 | NDCG@30 | MAP | MRR |
|---------|--------|------|-------|-------|--------|---------|---------|-----|-----|
| Gowalla | RMV | 0.1133 | 0.1500 | 0.0503 | 0.0839 | 0.0846 | 0.1053 | 0.0634 | 0.0636 |
| | DRE | 0.1333 | 0.2317 | 0.0532 | 0.1070 | 0.1036 | 0.1491 | 0.0499 | 0.0756 |
| | $k$-Medoids | 0.1500 | 0.1933 | 0.0669 | 0.1097 | 0.1136 | 0.1349 | 0.0859 | 0.0892 |
| | DPP | **0.1933** | **0.2433** | **0.0939** | **0.1427** | **0.1415** | **0.1734** | **0.1030** | **0.1146** |
| Amazon-Books | RMV | 0.2967 | 0.3633 | 0.2042 | 0.2473 | 0.2372 | 0.2590 | 0.1894 | 0.2096 |
| | DRE | 0.0483 | 0.1617 | 0.0307 | 0.0698 | 0.0674 | 0.1105 | 0.0246 | 0.0508 |
| | $k$-Medoids | 0.3200 | 0.3833 | 0.2225 | 0.2628 | 0.2624 | 0.2929 | 0.2128 | 0.2400 |
| | DPP | **0.3833** | **0.4567** | **0.2419** | **0.2968** | **0.3032** | **0.3353** | **0.2403** | **0.2730** |

questionnaire. Table 10 illustrates that the combination of a 50-item questionnaire along with a series of 5 dynamic 5-item questionnaires outperforms the 100-item questionnaire, which highlights the effectiveness of our PERE method.

Table 10: Comparison between a burn-in questionnaire using DPP and PERE with 100 elicited items for each method on Amazon-Books dataset.

| Datasets | Method | NDCG@10 ↑ | MRR ↑ |
|----------|--------|-----------|-------|
| Gowalla | Burn-in | 0.1497 | 0.1335 |
| - | PERE | **0.1806** | **0.1518** |
| Amazon-Books | Burn-in | 0.3388 | 0.3152 |
| - | PERE | **0.3616** | **0.3235** |

### F.3 GREEDY AND DPP COMPARISON

While the greedy method chooses the most popular item, we employ the Determinantal Point Process (DPP) in the 'burn-in' phase to achieve a better balance between diversity and popularity. DPP is advantageous in scenarios where preferences may diverge from mainstream popularity, ensuring a tailored and inclusive experience. Table 11 demonstrates that our method is more effective than the greedy method in constructing a personalized questionnaire for new users with 100 elicited items.

Table 11: Comparison between PERE and Greedy method on Amazon-Books dataset.

| Methods | NDCG@10 ↑ | MAP ↑ | MRR ↑ |
|---------|-----------|-------|-------|
| Greedy | 0.3415 | 0.198 | 0.3043 |
| PERE | **0.3616** | **0.2930** | **0.3235** |

## G MAIN EXPERIMENT SETTING

### G.1 DATASETS DESCRIPTION

In this paper, we use Gowalla [Cho et al., 2011] dataset and Amazon-Books [Ni et al., 2019] dataset. We report the statistics of Gowalla and Amazon-Books datasets in Table 12. The description for each dataset is the following:

- Gowalla is a location-based dataset that contains information about user check-ins at various locations.
- Amazon-Books is a subset of the Amazon Product Review dataset, specifically centered on book products. This dataset comprises reviews and user ratings for various products.

Amazon-Books includes both explicit and implicit user responses related to book products, whereas Gowalla exclusively provides implicit information indicating user preferences toward different locations. We employ two well-known methods to generate collaborative filtering embeddings for items: LightGCN and biVAE. LightGCN is trained solely to predict user-item

Table 12: Characterisitics of datasets used in our experiments.

| Dataset | Train User # | Item # | Interaction # | Density |
|---------|--------------|--------|---------------|---------|
| Gowalla | 28858 | 40981 | 1027370 | 0.00084 |
| Amazon-Books | 51643 | 91599 | 2984108 | 0.00062 |

interactions, making it suitable for datasets with implicit responses. On the other hand, biVAE is designed to predict specific ratings for user-item interactions, which necessitates explicit responses. Given that Gowalla contains only implicit responses, we exclusively use LightGCN on this dataset. However, since Amazon-Books contains explicit and implicit responses, we can utilize LightGCN and biVAE on this dataset.

### G.2 BASELINE DESCRIPTION

This paper uses a total of 7 baselines, which can be divided into fixed questionnaire generation methods and sequential questionnaire generation methods. Fixed questionnaire generation methods include:

- RMV: Please refer to Section F.2.
- $K$-medoids: Please refer to Section F.2.
- DRE: initially, this method defines a categorical distribution for sampling seed items from the entire item pool. Subsequently, it simultaneously learns the categorical distributions and a neural reconstruction network to infer users' preferences based on collaborative filtering (CF) information from the sampled seed items. Then, the encoder is utilized to select the seed items, while the decoder is used to recommend the favorite items.
- DPP: Please refer to Section C.

Sequential questionnaire generation methods include:

- PEO: This method presents a novel elicitation approach to construct a static preference questionnaire. It formulates the task of generating preference questionnaires, encompassing relative questions for new users as an optimization problem that can be solved in linear time of the number of items.
- Conditional DPP: Conditional DPP is a modified version of DPP that selects $K$ items from the remaining set of items.
- DPE: This preference elicitation model employs multi-armed bandits to diversify the seed item set through topic and item diversity.

### G.3 IMPLEMENTATION DETAILS

We use the standard codebase of LightGCN[1] and cornac implementation of biVAE[2] to generate item embedding and new user embedding. Afterward, we generate a new user according to Section 4 and use it as ground truth in our evaluation. This characteristics generation is necessary because we want to model experience probability that allows users to skip a question (NA response) in our questionnaire.

## H  INCONSISTENCY IN ELICITATION

In this section, we further introduce a method that can tweak the Chebyshev center to account for the inconsistent elicitation. Let $|\mathbb{P}|$ denote the cardinality of the set $\mathbb{P}$. Suppose we tolerate $\tau\%$ of inconsistency, i.e., at most $\tau|\mathbb{P}|$ preferences in the set $\mathbb{P}$ that can be violated. We define $\mathcal{U}_{\mathbb{P}}^{\tau}$ as the set of vectors $u_c$ with at most $\tau\%$ inconsistency with the preference set $\mathbb{P}$. This set can be represented using auxiliary binary variables as

$$\mathcal{U}_{\mathbb{P}}^{\tau} = \left\{ u \in \mathbb{H} : \begin{array}{l} \exists \gamma_{ij} \in \{0,1\} \ \forall v_i \succsim v_j \in \mathbb{P} \\ \sum_{(i,j)\in\mathbb{P}} \gamma_{ij} \leq \tau|\mathbb{P}| \\ 2u_c^{\top}(v_j - v_i) \leq \|v_j\|_2^2 - \|v_i\|_2^2 + \gamma_{ij}\mathbb{M} \end{array} \right\},$$

where $\mathbb{M}$ is a big-M constant. Intuitively, $\gamma_{ij}$ is an indicator variable: $\gamma_{ij} = 1$ implies that the preference is inconsistent.

---
[1] https://github.com/gusye1234/LightGCN-PyTorch
[2] https://github.com/recommenders-team/recommenders/tree/main

**Theorem 2** (Chebyshev center with inconsistent elicitation). *Given a tolerance parameter $\tau \in (0,1)$. The Chebyshev center $u_c^\star$ of the set $\mathcal{U}_\mathbb{P}$ can be found by solving the following problem*

$$
\begin{aligned}
\max \quad & r \\
\text{s.t.} \quad & 2u_c^\top(v_j - v_i) + 2r\|v_j - v_i\|_2 \leq \|v_j\|_2^2 - \|v_i\|_2^2 + \gamma_{ij}\mathbb{M} \; \forall v_i \succsim v_j \in \mathbb{P} \\
& \textstyle\sum_{(i,j)\in\mathbb{P}} \gamma_{ij} \leq \tau|\mathbb{P}| \\
& u_c \in \mathbb{H}, \; r \in \mathbb{R}_+, \; \gamma_{ij} \in \{0,1\} \; \forall v_i \succsim v_j \in \mathbb{P},
\end{aligned}
$$

*where $\mathbb{M}$ is a big-M constant.*

*Proof.* The optimization problem to find the Chebyshev center and its radius can be rewritten as

$$
\begin{aligned}
\max \quad & r \\
\text{s.t.} \quad & 2(u_c + \delta)^\top(v_j - v_i) \leq \|v_j\|_2^2 - \|v_i\|_2^2 + \gamma_{ij}\mathbb{M} \; \forall \delta \in \mathcal{B}_r, \; \forall v_i \succsim v_j \in \mathbb{P} \\
& \textstyle\sum_{(i,j)\in\mathbb{P}} \gamma_{ij} \leq \tau|\mathbb{P}| \\
& u_c \in \mathbb{H}, \; r \in \mathbb{R}_+, \; \gamma_{ij} \in \{0,1\} \; \forall v_i \succsim v_j \in \mathbb{P},
\end{aligned}
$$

where $\mathcal{B}_r = \{\delta \in \mathbb{R}^d : \|\delta\|_2 \leq r\}$ is a $d$-dimensional Euclidean ball of radius $r$. Pick any preference $v_i \succsim v_j \in \mathbb{P}$, the semi-infinite constraint

$$
2(u_c + \delta)^\top(v_j - v_i) \leq \|v_j\|_2^2 - \|v_i\|_2^2 + \gamma_{ij}\mathbb{M} \; \forall \delta \in \mathcal{B}_r
$$

is equivalent to the robust constraint

$$
2u_c^\top(v_j - v_i) + 2 \sup_{\|\delta\|_2 \leq r} \delta^\top(v_j - v_i) \leq \|v_j\|_2^2 - \|v_i\|_2^2 + \gamma_{ij}\mathbb{M}.
$$

Because the Euclidean norm is a self-dual norm, we have

$$
\sup_{\|\delta\|_2 \leq r} \delta^\top(v_j - v_i) = r\|v_j - v_i\|_2.
$$

Substituting the above relationship to the optimization problem completes the proof. $\qquad\square$