# OpenReview forum: "Cold-start Recommendation by Personalized Embedding Region Elicitation"
_auai.org/UAI/2024/Conference — UAI 2024 poster_

### Official Review · Reviewer_VH55 · 2024-03-05

**Q2-1 Originality-Novelty:** 2
**Q2-2 Correctness-Technical Quality:** 3
**Q2-5 Clarity Of Writing:** 2

**Q1 Summary And Contributions:**

The authors study the problem of cold-start recommendation, i.e., recommending an item to a user while having little to no information on the preferences of said user. To solve this problem, the authors propose a 2 steps method:
- A first part (burn-in) where the user has to declare for a set of K popular items (not necessarily the K most popular, but a mix between popularity and diversity) which items they like or dislike, with also the possibility to not declare if they like an item or not.
- A second part (sequential) where the user also have to rate items, but this time the items are presented sequentially, depending on the previously gathered preferences.

To determine which items are to be presented (or recommended) to the user, the authors submitted a distance-based method, in which the collected preferences delimit a region and the items within the region are then selected according to their distance to the Chebyshev center of said region. With each round of the sequential step, the region gets furthermore refined.

They illustrated their approach on different experiments that show that an improvement compared to other approaches in the state of the art.

**Q2-3 Extent To Which Claims Are Supported By Evidence:**

3: Good: the main claims are supported by convincing evidence (in the form of adequate experimental evaluation, proofs, (pseudo-)code, references, assumptions).

**Q2-4 Reproducibility:**

2: Fair: key resources (e.g. proofs, code, data) are unavailable but key details (e.g. proof sketches, experimental setup) are sufficiently well-described for an expert to confidently reproduce the main results.

**Q3 Main Strengths:**

The paper seems to be technically sound, with the assumptions clearly stated and highlighted (including the mathematical and probabilistic models used), which is very useful to determine both the strengths and the limitations of the work. Overall, I think the work of the authors makes a lot of sense in the field of personalized recommendation, and the proposed methods are justified. Their work are in my opinion in the scope of the conference, as we start with little information on the preferences, or even total uncertainty, needing some tools to gather knowledge and then make a decision.

A lot of interesting technical details and supports for the claims can be found both in the paper and in the supplementary material. Said supplementary contains a lot of useful information (which are, unfortunately, not always clearly linked in the main article).

**Q4 Main Weakness:**

I find the English in the paper mildly infuriating, as at the same time there are not so many typos and a clear effort, but also a lot of unwieldy sentences with old-fashioned/too formal for nothing words. While having a well-written formal paper is pleasant, I do believe that it should not be against the sake of clarity and simplicity, as most readers will not be native English-speakers. Overall, I think the clarity of the paper can be improved, especially as the paper has some interesting content to present.

Sadly, there is no link to the experiments made by the authors, but we still can find some details to reproduce ourselves the experiments. I think this point is extremely important for reproducibility, and I would prefer, if possible, a link to the experiments rather than having to make everything from scratches. I also think the conclusions of the experiments could be improved with some statistical tests (to determine whether PERE is really better than the other methods).

**Q5 Detailed Comments To The Authors:**

Paper:
- In general, when you link to the supplementary material, please say to which section of the said material you want use to lead to (A, B, C, etc.).
- When you start talking about DDP in "Contributions.", you could directly link to the supplementary material, as you do not describe outside of it what DDP really is (only a quick sentence to say what it does on the paper).
- I would suggest that the authors look at the works of Adam. L and Destercke. S (especially their 2021 article in UAI), the works of Bourdache. N and Perny P or in general the works of Boutillier C. about Current Solution Strategy in incremental elicitation, as they are likely related to their field. They also start with no information on the preferences of a user (thus a cold start) and they do some elicitation to recommend an alternative. They determine the region of possible parameters for an aggregation function that represents the preferences of a user (starting with no information on the preferences, thus usually a simplex). With incremental elicitation, the user typically has to compare pairs of alternatives, and such pairs of alternatives are selected each step according to the previous answers so that the set of possible parameters is refined as fast as possible with little cognitive effort (like your work). Moreover, the works of Adam and Bourdache deal with uncertainty coming from incoherent answers. Therefore, the novelty of the authors work is perhaps not as strong as said in the contributions, but their work is still interesting as I see many links but with a different approach, and can be justified through different points (previous works suppose an aggregation function on the preferences for example, and are not based on a distance metrics, here the users have to make assessments instead of comparing pairs of alternatives...).
- Do you think you can give more details on the parameter K in the assumption 2? What does the tolerance parameters mean? The user probably does not know this parameter (but perhaps has some insight on their tolerance).
- The authors clearly state in Assumption 3 that they do not allow for incoherent answers. However, what happens if the user start by giving wrong preferences? Do your method continues in a wrong direction, as incoherent answers are to be avoided? This reminds me a problem of CSS algorithm for incremental elicitation, where a wrong answer can lead to bad recommendations, and CSS algorithm will never question itself and continue in a wrong direction. Do you think the assumption 3 could be weakened to allow for incoherence detection/robustness?
- Why in the first phase we only pick among popular items, even for diversity purposes? Could we pick also niche items for diversity? How said popularity is measured?
- I am not totally convinced by the Sequential Q&A part of the Figure 1, as I don't find it so clear. I would perhaps show that for each step there are two sets (+1 and -1) and show at least 3 steps.
- Chebyshev center is a fine method to find a center for a polytope, but I wonder if you looked at other methods to find such a center?
- When do the Q&A stops? I have not seen any kind of metrics to tell when we have enough information to recommend an item. Speaking of which, 100 items seem extremely high to me! Would a real user really accept to rate 100 items?
- This work would clearly benefit from having statistical tests for each experiment to determine whether your method is statically more efficient than the others. Just a mean is not enough to take into account randomness/variability.
- I would love to see some experiments (especially plots) where we can see the performance of the different algorithms depending on some changes on hyperparameters. For example, I would love to know with how many items in total (for example 10 - 20 - 50 - 100) your method starts to be the most interesting/less interesting, etc.

English (some examples):
- What do you mean by a "Success element"?
- The expression "Fixed seed set of items" sounds very strange. Wouldn't "Fixed set of items" be enough?
- In the abstract, sometimes you say "User's preference" and sometimes "User's preferences". The second option should always be used.
- In the abstract "Adaptive items" does not make a lot of sense, as the items are not changed according to the preferences of the user, but their selection. Thus, I would write "Adaptive selection of items".
- Some words may be too formal (ascertain, delineates...) given they have some widely more used equivalents (determine, sketches...), but my main problem is with "wherein", that is old-fashioned/archaic and can simply be replaced by "in which".

**Q9 Complying With Reviewing Instructions:**

Yes

---

> ### Author Rebuttal · Authors · 2024-04-08
>
> We appreciate your review and insightful comments. We will incorporate a discussion of suggested related works in our revision. Note that we did provide the implementation code in the supplementary material.
>
> > $\kappa$ in Assumption. 2
>
> The tolerance parameter $\kappa$ in Assumption. 2 is a parameter controls how quickly the probability of a user experiencing an item decreases as the item moves further away from the user in the embedding space. A higher value of $\kappa$ implies that users are less tolerant of items that are dissimilar to their preferences, while a lower value suggests they are more open to diverse experiences.
>
>
>
>
> > Assumption 3
>
> We conducted two experiments to evaluate our method's performance under inconsistent user preferences:
>
> Exp 1: We introduce a probability $\tau$ that a user's response to an experienced item will be flipped, contradicting Assumption. 3. When $\tau=0$, there is no inconsistency, and when $\tau=1$, responses are always inconsistent. We plot the performance gain in NDCG@50 against the number of displayed items for different values of tau (https://anonymous.4open.science/r/recsys-4FD6/growth_NDCG_50.png). As tau increases, the performance gain decreases but remains positive, demonstrating that our method still provides benefits despite inconsistencies in user responses.
>
> Exp 2: We compare our method against DPE and RMV in the presence of inconsistency. The results, presented in the linked tables (https://anonymous.4open.science/r/recsys-4FD6/README.md#:~:text=2.%20Inconsistency%20experiments), show that our method maintains its advantage over the baselines even with inconsistent user preferences.
>
> We take a step further to address the reviewer’s concern thoroughly. Suppose that the user has a probability $\tau \in (0, 1)$ to flip their choice, we can tweak the Chebyshev center to account for this noise model. However, the new optimization problem becomes a mixed-binary optimization problem, which is significantly harder to solve and which requires an additional parameter $\tau$ that is difficult to tune. The formulation of the new Chebyshev center can be found here: https://anonymous.4open.science/r/recsys-4FD6/inconsistent_chebyshev.png.
>
> > Choosing popular items
>
> Item popularity is measured by user interactions based on historical data. As we assume that the probability of a user experiencing an item depends on its popularity and consider the data sparsity (we report the density of data in Table 10), we choose popular items to reduce the likelihood of N/A responses.
>
> > Figure 1
>
> In the Sequential Q&A phase, we keep track of two sets of items: $\mathcal{L}^+$ (liked), $\mathcal{L}^-$ (disliked) to generate all valid pairwise preferences by coupling items from the $\mathcal{L}^+$ and $\mathcal{L}^-$ sets. Each step is a Q&A round.
>
> > Methods to find polytope center
>
> There are other methods to find the polytope center, we compare them with the Chebyshev center as follows:
> Finding the Chebyshev center requires solving a linear optimization problem, while the analytic center requires solving a convex nonlinear optimization problem. Moreover, the Chebyshev center is invariant to redundant constraints, while the analytic center can be pushed near the boundary of the polytope by using redundant constraints [1].
> Finding a good approximation of P-center is computationally intensive. It requires a huge number of iterations just to reach near about centroid [1].
>
> > Size of the questionnaire
>
> In our questionnaire, we display selected items to the user, but we **do not require them to rate every item**. As illustrated in Figure 5, each item includes a "skip" option, which is equivalent to an N/A response in our choice model. This allows users to refuse to elicit their preference for an item or indicate that they have never experienced it. The "skip" option is a practical consideration that accounts for real-world user behavior.
>
> Although the total number of displayed items across the five rounds may seem large (50 items), our experiments show that users tend to rate only a small subset of these items. Specifically, in the Amazon-Books dataset, users rated an average of just 16.40 items out of the 100 displayed items (see Table 5 for more details).
>
> >  Statistical test
>
> We conduct additional statistical tests to study the effectiveness of our method. The detailed settings and results can be found at https://anonymous.4open.science/r/recsys-4FD6/README.md#:~:text=4.%20Varying%20K-,5.%20Statistical%20test,-NDCG%4010%3A.
>
> > Performance and total number of items
>
> We run an additional experiment in the rebuttal to study the impact of the total number of elicited items $K$ on two performance metrics: NDCG@10 and MRR. We report results at https://anonymous.4open.science/r/recsys-4FD6/README.md#:~:text=0.179-,4.%20Varying%20K,-5.%20Statistical%20test.
>
> **References**
>
> [1] Inayatullah et al,. "A New Technique for Determining Approximate Center of a Polytope".

---

### Official Review · Reviewer_W3rq · 2024-03-18

**Q2-1 Originality-Novelty:** 2
**Q2-2 Correctness-Technical Quality:** 3
**Q2-5 Clarity Of Writing:** 3

**Q1 Summary And Contributions:**

The paper studies the cold-start problem in recommendation systems. This is an important problem as learning about new users quickly allows recommendation systems to accurately curate items that suit the new users' taste, driving engagement and retention. The authors propose an algorithm with the goal of learning as much as possible about a new user by a combination of passively showing the user a set of pre-determined items as well as actively eliciting the user's feedback for an adaptively curated set of items. The authors propose a rating model based on user-item embedding distance. The main algorithm constructs a region where the true user embedding falls in and continually refines this region with adaptively chosen elicitations.

**Q2-3 Extent To Which Claims Are Supported By Evidence:**

3: Good: the main claims are supported by convincing evidence (in the form of adequate experimental evaluation, proofs, (pseudo-)code, references, assumptions).

**Q2-4 Reproducibility:**

3: Good: key resources (e.g. proofs, code, data) are available and key details (e.g. proofs, experimental setup) are sufficiently well-described for competent researchers to confidently reproduce the main results.

**Q3 Main Strengths:**

The paper explores a topic that has always been of significant interest in the recommendation systems research community. According to the authors "[their] method is the first to model new users using a set on the embedding space and computationally narrows the set with each cut after each round of questions". The paper is quite well written and that makes following the authors' argument easier. The experiment results do demonstrate that the proposed approach performs better than the baselines that been proposed in the literature.

**Q4 Main Weakness:**

1. The most important weakness I see in the paper is the strong assumption of consistency in the choice model. This is summarized in assumption 3 which states that preference follows a strict ordering as determined by the distance between the user and the items in embedding space. This is a very strong assumption while many embedding models are designed to specifically account for intransitivity in preference and comparison data such as the classical paper of Chen and Joachims [1]. The strong assumption seems to be used to facilitate the algorithm that performs cuts based on user-item like dislike data. Intuitively, every user-item feedback is used to remove narrow down the user embedding by isolating the half-space that the user's embedding falls in. The correctness of this approach seems to rely crucially on Assumption 3.

2. In my opinion, the authors have proposed a model without a convincing justification as to why the model is significantly better than the existing model at explaining real life data. The authors claim in the last paragraph of Section 5 that the model improves over existing model by explicitly modeling the probability that a user has already experienced an item and therefore not providing feedback (user gives NA rating). However, this seems like a minor innovation and is used to ensure the consistency assumption (Assumption 3) holds. On a related note, It’s not clear to me why the user-item experience probability in (3) and (4) are well motivated.

3. The experiments are not very convincing in the sense that the set up is semi-synthetic where the user and item embedding are first obtained using another collaborative filtering method (for e.g., LightGCN). The embedding is then used to generate the data following the proposed model in (3) and (4). The different algorithms are then compared on this semi-synthetic data. It is therefore not at all surprising that the active learning method does better than the passive methods.

4. The proposed method is largely a heuristic as opposed to a theoretically well founded algorithm based on a well defined model. The theoretical assumptions of the model are strong (see point number 1). This means that it should be possible to obtain some guarantee in terms of the number of elicitations needed in order to accurately (or approximately) estimate the user’s embedding.

5. A quick google research returned the paper [2] which looks at cold-start item recommendations. While the paper looks at learning the representation of a new item, item and user are in some sense interchangeable. Since a lot of the metrics presented in the paper are also covered by [2], it seems natural to ask why you didn't compare your algorithm with [2].

[1] Modeling Intransitivity in Matchup and Comparison Data, Chen and Joachims.
[2] Item cold-start recommendations: learning local collective embeddings, Saveski and Mantrach

**Q5 Detailed Comments To The Authors:**

1. It's not entirely clear to me how useful or realistic the choice model, essentially captured by Assumption 2 and especially Assumption 3, really is. The preference consistency assumption just seems like a really strong assumption that is needed to ensure the correctness of the embedding region finding algorithm.

2. Why did you only look at these 2 particular datasets. The framework seems general enough that one can look at multiple other datasets such as those considered in the papers listed in the related work section?

**Q9 Complying With Reviewing Instructions:**

Yes

---

> ### Author Rebuttal · Authors · 2024-04-08
>
> Dear reviewer,
>
> We thank you for the review and thoughtful feedback! We will include these discussions in the revision for clarity. We address your concerns as follows:
>
> > Q4.1 + Q5.1
>
> The reviewer might have misunderstood the implication of Assumption 3 in our paper.
>
> Our ultimate goal is to locate embedding values for a new user from the preference elicitation framework. There can be many possible embedding values that can induce the same instance of preference data. If the user can give noisy feedbacks (aka, have an inconsistent preference), the set of possible embedding values is even bigger!
>
> Assumption 3 does not restrict that the user must have consistent preferences. Asm. 3 simply guides us that: given the preference data, we will focus on the smaller set of embedding values that correspond to consistent preferences. Asm. 3 can be considered an inductive bias to make the search space smaller, thus more computationally tractable. As an analogy, many ML models are linear models, and linear models perform competitively in reality despite the fact that data can be generated from nonlinear processes.
> Despite focusing on a smaller set of embedding values, we show in Tab. 1-3 of our main paper that our method is better than the baselines.
>
> The reviewer may be tempted to drop Assumption 3, but the reviewer will need to introduce a noise model to account for inconsistencies in the user responses. This requires introducing yet another strong assumption on the noise model, which is also hard to justify/verify in practice (for example, the reviewer may need to assume a Bradley-Terry-Luce choice model, etc.). Moreover, accounting for noisy responses will definitely increase the computational complexity of the problem.
>
> The reviewer, however, brought up a relevant point that we have not benchmarked our method under noisy data. To address this concern, we conducted two experiments to evaluate our method's performance under inconsistent user preferences:
> Exp 1: We introduce a probability $\tau$ that a user's response to an experienced item will be flipped. When $\tau = 0$, there is no inconsistency, and when $\tau = 1$, responses are always inconsistent. We plot the performance gain in NDCG@50 against the number of displayed items for different values of $\tau$ (https://anonymous.4open.science/r/recsys-4FD6/growth_NDCG_50.png). As tau increases, the performance gain decreases but remains positive, demonstrating that our method still provides benefits despite inconsistencies in user responses.
>
> Exp 2: We compare our method against DPE and RMV in the presence of inconsistency. The results (https://anonymous.4open.science/r/recsys-4FD6/README.md#:~:text=3.%20Inconsistency%20Experiment) show that our method maintains its advantage over the baselines even with inconsistent user preferences.
>
> We take a step further to address the reviewer’s concern thoroughly. Suppose that the user has a probability $\tau \in (0, 1)$ to flip their choice, we can tweak the Chebyshev center to account for this noise model. However, the new optimization problem becomes a mixed-binary optimization problem, which is significantly harder to solve and which requires an additional parameter $\tau$ that is difficult to tune. The formulation of the new Chebyshev center can be found here: https://anonymous.4open.science/r/recsys-4FD6/inconsistent_chebyshev.png
>
> In summary, Assumption 3 does not imply that our method is unable to handle inconsistencies in user responses. Our experiments demonstrate that our preference elicitation method remains effective even when the users give noisy responses. We hope this clarifies the role of Assumption 3 in our work and addresses the reviewer's concerns.
>
> > Q4.2
>
> We model the experience probability as a product of the normalized item popularity score and the estimated relevance score of the item to the user. The first term gives higher weights to more popular items. The second term captures the intuition that items closer to the user's preferences are more likely to have been experienced [2].
>
> > Q4.3. + Q5.2
>
> To address this, we have conducted additional experiments on three real-world large-scale datasets. The detailed settings and results of these experiments can be found in Section 1: https://anonymous.4open.science/r/recsys-4FD6/README.m. Our method outperforms baselines across these real-world datasets.
>
> > Q4.5
>
> While Saveski and Mantrach and our work share the same goal, the two approaches differ significantly, making a direct comparison unsuitable. Our method relies solely on item embeddings. We do not require access to additional item properties or user information. In contrast, [1] needs extra item properties to learn the mapping from content of new item / user to the latent space.
>
> **References**
>
> [1] Item cold-start recommendations: learning local collective embeddings, Saveski and Mantrach.
>
> [2] Neural graph collaborative filtering. Wang, X., He, X., Wang, M., Feng, F. and Chua, T.S.

---

### Official Review · Reviewer_7AnH · 2024-03-22

**Q2-1 Originality-Novelty:** 2
**Q2-2 Correctness-Technical Quality:** 2
**Q2-5 Clarity Of Writing:** 2

**Q1 Summary And Contributions:**

The paper addresses the cold-start problem for new users in recommendation systems by proposing a two-stage personalized rating inquiry scheme. Initially, it gauges user preferences by asking for ratings on popular items; then, it refines these preferences through sequential inquiries. Throughout, user preferences are treated as a range, not a single point, for more accurately capturing interests. The paper validates this approach with experiments, showing its effectiveness in improving recommendations for new users.

**Q2-3 Extent To Which Claims Are Supported By Evidence:**

2: Fair: the main claims are somewhat supported by evidence (but the experimental evaluation may be weak, or does not match entirely with the claims, important baselines may be missing, proofs contain important ideas but lack rigor, algorithmic details are only discussed superficially, references are imprecise, assumptions are not sufficiently motivated or explicated, etc.).

**Q2-4 Reproducibility:**

2: Fair: key resources (e.g. proofs, code, data) are unavailable but key details (e.g. proof sketches, experimental setup) are sufficiently well-described for an expert to confidently reproduce the main results.

**Q3 Main Strengths:**

1.Unlike traditional methods that use point estimates to represent user preferences, this approach employs interval estimation to more comprehensively capture the uncertainty in user preferences.
2.The two-stage inquiry strategy enables personalized adaptation to the unique preferences of different new users
3.Relatively well-structured and well-formatted paper.

**Q4 Main Weakness:**

1.While the personalized rating inquiry strategy can improve recommendation quality, it may also increase the burden on new users, especially during the initial phase that requires rating multiple items, raising concerns about its practicality.
2. Lacks reporting on the model's time and space complexity.
3. While the method's effectiveness on specific datasets is validated, there is a lack of discussion on its generalizability across different domains and types of recommendation systems.
4. There is a shortage of intuitive theoretical derivations to justify the model's design and a lack of visual demonstrations or data analysis on how the model affects recommendation results.

**Q5 Detailed Comments To The Authors:**

1.How do you view the potential issue of the proposed method increasing the burden on new users? Have you considered strategies to simplify the rating inquiry process to alleviate user burden?
2.Could you elaborate on the complexity and computational cost of the algorithm? How would performance and efficiency be affected when applied to large-scale datasets?
3.How do you evaluate the strengths and weaknesses of your proposed method compared to current rating inquiry and cold-start solutions? Could you provide more comprehensive comparative experiments that also consider factors like time performance?
4.Could you provide more discussion on the limitations of this method and directions for future development?

**Q9 Complying With Reviewing Instructions:**

Yes

---

> ### Author Rebuttal · Authors · 2024-04-08
>
> Dear reviewer,
>
> We thank you for the review and thoughtful feedback! We address your concerns as follows:
>
> > Q4.1 + Q5.1. Practicality of our method
>
> There seems to be a misunderstanding regarding our problem setting and the requirements for user input.
>
> In our questionnaire, we display selected items to the user, but we **do not require them to rate every item**. As illustrated in Figure 5 of the Appendix, each item in the questionnaire includes a "skip" option, which is equivalent to an N/A response in our choice model. This allows users to refuse to elicit their preference for an item or indicate that they have never experienced it. The "skip" option is a practical consideration that accounts for real-world user behavior and preferences.
>
> Although the total number of displayed items across the five rounds may seem large (50 items), our experiments show that users tend to rate only a small subset of these items. Specifically, in the Amazon-Books dataset, users rated an average of just 16.40 items out of the 100 displayed items (see Table 5 in the Appendix for more details). This indicates that the actual rating burden on users is much lower than it might appear at first glance.
>
> > Q4.2.  +  Q5.2. Time and space complexity of our model
>
> DPP [1]: $O((k^2) * N)$ in time complexity and $O(N^2)$ in storing kernel matrix
>
> Sequential Q&A: $O(N * (|\mathcal{L}^+| + |\mathcal{L}^-|))$ in time complexity. In practice, $|\mathcal{L}^+|$ and $|\mathcal{L}^-|$ are quite small (as shown in Table 5 in our appendix). The space complexity of Sequential Q&A is $O(N * d)$.
>
> Finding Chebyshev Center: This step involves solving a linear programming problem. We use Gurobi [2] to solve this problem and running time is negligible compared to other phases.
>
> where $N$ is the number of popular items, k is recommendation size, $|\mathcal{L}^+|$ and $|\mathcal{L}^-|$ are liked/disliked items size, and $d$ is dimension of embeddings.
>
> We provide numerical experiments on Amazon-Books (a relatively large-scale dataset) to measure time performance in Q5.3.
>
> > Q4.3. Generalization of our method across domains and recommender systems
> Our method is designed to be generalizable across different domains and types of recommendation systems, as long as well-trained item embeddings are available. The proposed framework is not limited to specific datasets or domains but rather relies on the quality of the item embeddings.
>
> We have validated the performance of our method against datasets from different domains, including Movielens (movies), Amazon Books (books), and Gowalla (location-based social networking). Please also refer to this link for additional experiments: https://anonymous.4open.science/r/recsys-4FD6/README.md
>
>
> > Q4.4. Impact on recommendation results
>
> To better understand how our model affects recommendation results, we plot the NDCG metric score against the number of displayed items (https://anonymous.4open.science/r/recsys-4FD6/README.md#:~:text=0.179-,4.%20Varying%20K,-5.%20Statistical%20test) of our method against baselines.  With an increasing number of displayed items, our preference elicitation framework improves the average recommendation quality to new users.
>
>
> > Q5.3. Strengths, weaknesses, and time comparisons
>
> Strength:
> Given a recommendation system with pre-trained item embeddings, we can quickly estimate embeddings of new users through a simple static & sequential questionnaire.
> Our method does not require any neural network training like in DRE [3], hence we do not suffer from generalizability or explainability issues.
> We consider an additional case where users just ‘skip' the questions rather than forcing users to answer their preferences for every item.
>
> Weakness:
> Slightly higher runtime than methods using neural networks or static questionnaires, but still reasonable in practice (avg. 0.272s per round, see table 1, section 6: https://anonymous.4open.science/r/recsys-4FD6/README.md#:~:text=0.039-,6.%20Time%20comparison,-Amazon%2DBooks.)
>
> > Q5.4.  Limitations of this method and directions for future development
>
> Currently, we do not leverage rich auxiliary information like user demographics, which could help warm-start the initial embedding. Extending our approach to refine an embedding region initialized with such data is a promising direction.
> As suggested by Reviewer GAdu, collecting a small seed set of responses via our framework to enable few-shot learning for new users is another interesting avenue for future exploration.
>
> **References**
>
> [1] Chen, L., Zhang, G. and Zhou, E., 2018. Fast greedy map inference for determinantal point process to improve recommendation diversity. Advances in Neural Information Processing Systems, 31.
>
> [2] Gurobi Optimization, L.L.C., 2021. Gurobi optimizer reference manual.
>
> [3] Kweon, Wonbin, et al. "Deep rating elicitation for new users in collaborative filtering." Proceedings of The Web Conference 2020. 2020.

---

### Official Review · Reviewer_GAdu · 2024-03-23

**Q2-1 Originality-Novelty:** 3
**Q2-2 Correctness-Technical Quality:** 3
**Q2-5 Clarity Of Writing:** 4

**Q1 Summary And Contributions:**

### Summary
This paper presented an interesting framework for preference elicitation of new users given item embeddings are known. The preference elicitation framework consists of 2 phases, one is the “burn-in” phase for new users to rate on popular items, and another one is to rate adaptive items to further refine the user embedding. It aims at solving the new user embedding generation problem, with questionaires to collect more user feedback and further infer new user embedding.

### Contributions
1. A novel framework for new user embedding generation.
2. Theoretical analysis to demonstrate the underlying mechanism behind the refinement process.
3. Experiments to analyze multiple aspects of the proposed framework.

**Q2-3 Extent To Which Claims Are Supported By Evidence:**

3: Good: the main claims are supported by convincing evidence (in the form of adequate experimental evaluation, proofs, (pseudo-)code, references, assumptions).

**Q2-4 Reproducibility:**

3: Good: key resources (e.g. proofs, code, data) are available and key details (e.g. proofs, experimental setup) are sufficiently well-described for competent researchers to confidently reproduce the main results.

**Q3 Main Strengths:**

1. Good writing and clear motivations.
2. The investigated problem is indeed important to the recsys community.
3. The experiments are reasonable and cover necessary analysis of components, after reading the appendix.

**Q4 Main Weakness:**

1. I have concerns on if the questionnaire way is a practical way to get more user feedback. In the paper, authors investigated 5 rounds, with 10 items in each round. This at least requires each new user to rate 50 items. Practically, most users in recsys do not even rate more than 10 items. This questionnaire way seems like an idealistic setting.
2. There are some inductive collaborative filtering ways to infer user embeddings for new users with additional new user feedback, such as the user rated items from the questionnaire. These literatures should be discussed because the questionnaire data can be used to directly infer user embedding.

[1]. Wu, Qitian, et al. "Towards open-world recommendation: An inductive model-based collaborative filtering approach." International Conference on Machine Learning. PMLR, 2021.

**Q5 Detailed Comments To The Authors:**

Please check the weakness part.

**Q9 Complying With Reviewing Instructions:**

Yes

---

> ### Author Rebuttal · Authors · 2024-04-08
>
> Dear reviewer,
>
> We appreciate that you review our paper and provide insightful comments. We would like to address your concerns as follows:
>
> > Q4.1.  I have concerns on if the questionnaire way is a practical way to get more user feedback. In the paper, authors investigated 5 rounds, with 10 items in each round. This at least requires each new user to rate 50 items. Practically, most users in recsys do not even rate more than 10 items. This questionnaire way seems like an idealistic setting.
>
> We would like to clarify that there seems to be a misunderstanding regarding our problem setting and the requirements for user input.
>
> In our questionnaire, we display selected items to the user, but we do *not require them to rate every item*. As illustrated in Figure 5 of the Appendix, each item in the questionnaire includes a "skip" option, which is equivalent to an N/A response in our choice model. This allows users to refuse to elicit their preference for an item or indicate that they have never experienced it. The "skip" option is a practical consideration that accounts for real-world user behavior and preferences.
>
> Although the total number of displayed items across the five rounds may seem large (50 items), our experiments show that users tend to rate only a small subset of these items. Specifically, in the Amazon-Books dataset, users rated an average of just 16.40 items out of the 100 displayed items (see Table 5 in the Appendix for more details). This indicates that the actual rating burden on users is much lower than it might appear at first glance.
>
> We hope this clarification addresses your concerns about the practicality of our questionnaire approach. By providing users with the flexibility to skip items and only rate a subset of the displayed options, we aim to strike a balance between gathering valuable feedback and minimizing user effort.
>
>
> > Q4.2. There are some inductive collaborative filtering ways to infer user embeddings for new users with additional new user feedback, such as the user rated items from the questionnaire. These literatures should be discussed because the questionnaire data can be used to directly infer user embedding.
>
> Thank you for suggesting the paper by Qitian et al. [1], which presents an interesting approach to handling few-shot and cold-start user recommendation problems using inductive collaborative filtering (IDCF). While their work is relevant to our research, there are some key differences between our settings and objectives.
>
> In [1], the authors assume that a small number of user interactions (few-shot data) are available for new users, which they use to inductively learn user embeddings. In contrast, our work focuses on designing preference elicitation algorithms to actively gather responses from new users, starting from a cold state without any prior interaction data.
>
> However, we recognize that combining our preference elicitation approach with the IDCF framework could lead to a powerful hybrid system. We could first collect a small number of user responses using our method and then employ the IDCF framework to generate more refined user embeddings.
>
> We will incorporate a discussion of [1] and other relevant inductive collaborative filtering methods in our related work section. This will provide a more comprehensive overview of the current state-of-the-art in handling new users in recommendation systems and highlight the potential for future research in this area.
>
> We thank the reviewer for raising such interesting discussions and constructive feedback and we hope that our answers adequately address your concerns. We will include these discussions in the revision for clarity. We are open to answering any remaining concerns the reviewer might have.
>
> **References**
>
> [1] Wu, Qitian, et al. "Towards open-world recommendation: An inductive model-based collaborative filtering approach." International Conference on Machine Learning. PMLR, 2021.

---

### Official Review · Reviewer_7bwK · 2024-03-27

**Q2-1 Originality-Novelty:** 3
**Q2-2 Correctness-Technical Quality:** 3
**Q2-5 Clarity Of Writing:** 3

**Q1 Summary And Contributions:**

This paper presents an approach to dealing with the long-standing cold-start problem in recommender systems. A two-phased personalised elicitation scheme is proposed. Firstly, a burn-in phase elicits seed preferences that can be used in a subsequent phase that involves a sequential Q&A.

**Q2-3 Extent To Which Claims Are Supported By Evidence:**

3: Good: the main claims are supported by convincing evidence (in the form of adequate experimental evaluation, proofs, (pseudo-)code, references, assumptions).

**Q2-4 Reproducibility:**

3: Good: key resources (e.g. proofs, code, data) are available and key details (e.g. proofs, experimental setup) are sufficiently well-described for competent researchers to confidently reproduce the main results.

**Q3 Main Strengths:**

The paper addresses a well-recognised problem in recommender system research and presents some supporting experiments.

**Q4 Main Weakness:**

No trial with real users is presented, which I believe would have added to the significance of the result.

**Q5 Detailed Comments To The Authors:**

This is a very nice paper addressing an important problem. While the results are good, and probably sufficient as they are, it will be important to undertake a user-study in subsequent work to really demonstrate the practical value of the approach.

**Q9 Complying With Reviewing Instructions:**

Yes

---

> ### Author Rebuttal · Authors · 2024-04-08
>
> Dear reviewer,
>
> We appreciate the reviewer's positive feedback and acknowledgment of the importance of the problem we addressed.
>
> > Q4.1. Weakness 1: No trial with real users is presented, which I believe would have added to the significance of the result.
>
> Regarding the lack of real user trials, we agree that evaluating our system with real users would add significant value to the paper. However, conducting an unbiased online evaluation requires access to a large user base, which we currently do not have. To mitigate this limitation and provide meaningful results, we designed additional offline experiments using real user data accumulated in datasets such as MovieLens 10M, MovieLens 20M, and Amazon Books.
>
> In these experiments, the items preferred by users are based on their actual ratings from the datasets, rather than being generated from the embeddings. We followed the data processing approach used in the recent Deep rating elicitation paper [1], which is also one of our baselines. Specifically, we filtered out users who rated over 40 items and converted implicit ratings (1-5) to explicit ratings (0 and 1) as follows: ratings of 4 and 5 are considered as liked items, while ratings of 0, 1, and 3 are considered as disliked items.
>
> Our preference elicitation method consistently outperforms the baseline methods across these real-world datasets. The performance metrics used to evaluate our method and the baselines include NDCG, MAP and MRR. The detailed results of these experiments can be found in the supplementary material (3 Tables in Section 1.): https://anonymous.4open.science/r/recsys-4FD6/README.md#:~:text=1.%20Real%2Dworld%20experiments.
>
> While offline evaluations cannot perfectly replicate online user behavior, we believe these experiments provide valuable insights into the effectiveness of our approach when applied to real-world data. We hope these additional results address the reviewer's concern about the lack of real user trials to some extent.
>
> We thank the reviewer for raising such interesting discussions and constructive feedback and we hope that our answers adequately address your concerns. We will include these discussions in the revision for clarity. We are open to answering any remaining concerns the reviewer might have.
>
> **References**
>
> [1] Kweon, Wonbin, et al. "Deep rating elicitation for new users in collaborative filtering." Proceedings of The Web Conference 2020. 2020.

---

### Meta-Review · Area_Chair_MzYZ · 2024-04-15

This paper tackles an important problem, that of preference elicitation for recommenders with an emphasis on cold-start. It brings together a collection of useful and appropriate techniques and present some reasonable empirical evaluation. All of the reviewers were positive on the paper to varying degrees. That said, the reviewers posed some questions and made several suggestions that should be used to improve the paper.

On my reading of the paper, I believe it would make a valuable contribution to UAI, except for one issue that I believe the authors should be able to fix. The paper seems to claim much more novelty than it actually exhibits, in part, because it does a very poor job of comparing and/or discussing wide swatches of related literature. First, it ignores the body of literature on active collaborative filtering that dates back many years, including at this conference; for example:
* Rashid, et al. IUI-2002. Getting to know you: Learning new user preferences in recommender systems.
* Boutilier, Zemel, Marlin, UAI-2003. Active collaborative filtering.
* Jin, Si, UAI-2004. A Bayesian Approach toward Active Learning for Collaborative Filtering.
* And much more...

The massive body of work on preference elicitation is also overlooked. Bayesian methods for recommenders that work in embedding space are certainly relevant, e.g.,
* Vendrov et al. AAAI-20. Gradient-Based Optimization for Bayesian Preference Elicitation.
More general Bayesian and entropy-based PE (not necessarily for collaborative-fltering style recommenders) methods should be contrasted briefly.

More importantly, the region-based approach taken here is remarkably similar in many respects to region-based and polyhedral methods for PE. Some use notions such as maximin or minimax regret to drive elicitation (different than here, but should be contrasted). Other methods use very similar methods to those proposed here (e.g., the analytic center methods used in conjoint analysis), e.g.,
* Toubia, Hauser, Simester, J. Marketing Research (2004). Polyhedral methods for adaptive choice-based conjoint analysis.
* Iyengar, Lee, Campbell, ACM-EC-2001. Q-Eval: Evaluating multiple attribute items using queries.
* And others.

The authors must discuss how their work differs from (and is similar) to some representative work from each of these areas.